# Learning to Think: Information-Theoretic Reinforcement Fine-Tuning for LLMs

**Jingyao Wang**[1,2]*, **Wenwen Qiang**[1,2]*†, **Zeen Song**[1,2]*, **Changwen Zheng**[1,2], **Hui Xiong**[3,4]

[1]Institute of Software Chinese Academy of Sciences, [2]University of Chinese Academy of Sciences,
[3]The Hong Kong University of Science and Technology (Guangzhou),
[4]The Hong Kong University of Science and Technology
{wangjingyao2023, qiangwenwen, songzeen}@iscas.ac.cn, xionghui@ust.hk

## Abstract

Large language models (LLMs) excel at complex tasks thanks to advances in their reasoning abilities. However, existing methods overlook the trade-off between reasoning effectiveness and efficiency, often encouraging unnecessarily long reasoning chains and wasting tokens. To address this, we propose Learning to Think (L2T) [3], an information-theoretic reinforcement fine-tuning framework for LLMs to make the models achieve optimal reasoning with fewer tokens. Specifically, L2T treats each query-response interaction as a hierarchical session of multiple episodes and proposes a universal dense process reward, i.e., quantifies the episode-wise information gain in parameters, requiring no extra annotations or task-specific evaluators. We propose a method to quickly estimate this reward based on PAC-Bayes bounds and the Fisher information matrix. Theoretical analyses show that it significantly reduces computational complexity with high estimation accuracy. By immediately rewarding each episode's contribution and penalizing excessive updates, L2T optimizes the model via reinforcement learning to maximize the use of each episode and achieve effective updates. Empirical results on various reasoning benchmarks and base models demonstrate the advantage of L2T across different tasks, boosting both reasoning effectiveness and efficiency.

## 1 Introduction

Large Language Models (LLMs) have progressed from handling basic natural language processing tasks to tackling complex problems, such as writing and maintaining code bases [23, 67, 38, 48], navigating the web and controling devices [66, 54, 17, 4], and acting as personal assistants [21, 30, 22, 7], thanks to the advances in their reasoning abilities. Recent results [24, 19, 56, 43] in LLM reasoning show that scaling test-time compute can substantially improve reasoning capabilities, e.g., [35, 5] demonstrated that generating more tokens during inference yields logarithmic-linear gains. Based on this, a new class of reasoning models [43, 53, 9, 49] has coupled test-time compute scaling with reinforcement learning (RL), achieving state-of-the-art (SOTA) results on various challenging benchmarks [15, 61, 16]. These models employ chain-of-thought (CoT) tokens to guide multi-step reasoning and maintain logical consistency throughout the solution process [50, 52, 58]; by extending and optimizing CoT paths to produce trajectories longer than typical correct solutions, they more thoroughly explore the solution space and thereby boost final answer accuracy [43, 35, 20].

Despite existing methods having demonstrated great performance, they still struggle to balance reasoning effectiveness and efficiency. Specifically, existing approaches typically rely on final

---

*Equal contribution.
†Corresponding author.
[3]Project page: https://wangjingyao07.github.io/L2T.github.io/

39th Conference on Neural Information Processing Systems (NeurIPS 2025).

outcome rewards for policy optimization, providing no feedback on intermediate reasoning steps. Under such delayed feedback, extending the reasoning chain does not incur any cost, and even a tiny accuracy gain from a large amount of extra reasoning steps is treated as a positive signal [59, 55]. Consequently, the models favor a "one more thought" and continually lengthen their CoTs, resulting in redundant computation and thus reducing reasoning efficiency. Our experiments in **Subsection 3.2** further demonstrate this (**Figure 1**): existing outcome-reward-based RL methods often lead LLMs to consume more than twice the tokens actually needed for the correct answer. Furthermore, by evaluating across different reasoning tasks, we find that this redundancy not only wastes resources but sometimes degrades reasoning effectiveness. For example, on difficult questions (e.g., Tier 4 multi-stage math questions [13]), moderate chain extensions improve coverage of critical steps; whereas on simple tasks (e.g., Tier 1 question "12 + 5"), overly long reasoning chains may reduce overall accuracy. Since real-world tasks vary, no fixed chain length is optimal for all cases. **Therefore, designing effective dense process rewards to assess the contribution of each reasoning step is both necessary and valuable. Such rewards help the model to generate tokens that most benefit the answer, ensuring reasoning effectiveness with minimal token budget and efficient learning.**

To this end, we propose Learning to Think (L2T), an information-theoretic reinforcement fine-tuning framework for LLMs. At its core, L2T proposes a universal information-theoretic dense process reward, which quantifies the information gain in model parameters. The proposed reward consists of (i) a fitting information gain term that drives the model to capture correctness-critical information in each update; and (ii) a compression penalty that discourages overly optimization, further preserving efficiency. By treating each question-answer pair as a session of multiple episodes and immediately rewarding each episode, it makes the model focuses on the process progress, thus curbing redundant reasoning steps and the resulting computational waste. This reward is independent of input format, label type, or task domain, and no extra annotations are needed. We leverage this reward to train the LLM (also the policy) via reinforcement learning to make it generate the tokens that best contribute to the answer correctness at each reasoning step. Specifically, L2T includes three stages: (i) Problem reformulation (**Subsection 4.1**): we treat each question-answer interaction as a hierarchical session of multiple episodes, where each episode represents a segment of the reasoning chain that underpins dense reward calculation and optimization; (ii) Reward design (**Subsection 4.2**): upon episode completion, we calculate the information-theoretic reward via PAC-Bayes bounds and the Fisher information matrix. Based on this, we halt unproductive reasoning and thus balance effectiveness and efficiency; (iii) LLM fine-tuning (**Subsection 4.3**): we optimize the LLMs by maximizing cumulative reward across tasks via reinforcement learning, ensuring reasoning effectiveness and efficiency.

Empirically, across challenging reasoning benchmarks (e.g., AIME, AMC, HumanEval) and base models (e.g., DeepScaleR-1.5B-Preview, DeepSeek-R1-Distill-Qwen-1.5B, DeepSeekR1-Distill-Qwen-7B), L2T consistently achieves comparable performance and stable improvements (**Section 5**). Compared to standard outcome-reward approaches (e.g., GRPO), it boosts performance by about 3.7% and doubles token efficiency; compared to process-reward baselines (e.g., ReST-MCTS, MRT), it raises accuracy by about 2% and increases efficiency by about 1.3$\times$. In multi-task evaluations, L2T delivers an average accuracy gain of nearly 3% across tasks of varying difficulty and maintains stable improvements under different token budgets. These results demonstrate the advantages of L2T, which effectively balances reasoning effectiveness and efficiency across diverse reasoning scenarios.

**The main contributions are as follows:** (i) We explore the trade-off between reasoning effectiveness and efficiency and propose Learning to Think (L2T), an information-theoretic reinforcement fine-tuning framework for LLMs. L2T decomposes each interaction into successive episodes and proposes a dense process reward to quantify each episode's performance gain; by optimizing LLM via episodic RL, it adaptively allocates reasoning depth across different tasks, enabling effective reasoning with limited but sufficient token budgets. (ii) We propose a universal information-theoretic process reward based on internal model signals, i.e., the information gain in model parameters, eliminating the need for external annotations or specialized evaluators. Leveraging PAC-Bayes bounds and the Fisher information matrix, we derive a scalable approximation of the intractable information gain with theoretical guarantees. (iii) Across diverse complex reasoning benchmarks and base models, L2T consistently achieves great performance, delivering boosts in both effectiveness and efficiency.

## 2 Related Work

**Reasoning of LLMs**   Complex reasoning has long been recognized as one of the most challenging capabilities for LLMs [23, 67, 19, 56, 43]. To enhance inference, several works [43, 53, 49, 57] have incorporated outcome-reward RL during fine-tuning. RL paradigm mainly extends and optimizes CoT paths based on test-time compute scaling to more thoroughly explore the solution space [35, 5]. However, recent studies [59, 9, 55, 5] show that reasoning length and accuracy are not strictly positively correlated. Excessively long CoTs not only consume undue tokens but also degrade performance (also demonstrated in **Subsection 3.2**). The wasting token budget reduces the efficiency of unit tokens under a limited budget due to problems such as attention dilution and context truncation, so that the final accuracy may decline [31, 44]. Some methods attempt to reduce reasoning depth via process rewards [63, 36], heuristic scoring [11, 20], or length penalties [57, 1, 33]. These approaches, however, require task-specific evaluators, incurring prohibitive annotation costs with poor cross-task reuse (with more comparison in **Appendices D and G**). To address these limitations, we propose a universal information-theoretic dense process reward and leverage reinforcement fine-tuning to adaptively recognize reasoning depth across diverse tasks. This design achieves the trade-off between reasoning effectiveness and efficiency without additional annotations or specialized evaluators.

**Process Reward Models**   Unlike outcome rewards that assess the final answer, a PRM evaluates the correctness of each intermediate reasoning step [29, 47]. PRMs can be trained via automated supervision without costly human-annotated process labels [34, 39, 60]. Once learned, a PRM can both guide test-time search by allocating additional compute and accelerate exploration in RL using the policy's own trajectories [43, 35, 60]. Recent works [42, 65, 63, 10, 64] have applied process rewards to fine-tune LLMs with RL, for example, [65] uses relative progress estimation to generate high-quality intermediate supervision labels; [57] derives exploration bonuses from length penalties or an LLM-based judge; and [36] introduces a regret-minimization process reward to optimize test-time compute. However, these methods depend on external annotations or task-specific evaluators, which are expensive to produce and difficult to reuse when task requirements change. In contrast, our approach uses internal model signals, i.e., information gain, as intermediate rewards to optimize the policy without additional interaction data. By constructing a universal dense process reward, our framework applies seamlessly across diverse tasks and promotes efficient, step-wise reasoning.

## 3 Problem Settings and Analysis

### 3.1 Problem Settings

Our goal is to fine-tune an LLM so that it can answer arbitrary questions $x$ within a fixed token budget $B_{\text{token}}$, enforcing efficient reasoning. We treat the LLM as a stochastic policy $\pi_\theta(\cdot \mid x)$ parameterized by $\theta$. Each question $x$ is drawn from an underlying distribution $\mathcal{P}_x$. Under the token budget $B_{\text{token}}$, the model may generate up to $T_{max}$ tokens at test time, and yields an output sequence $z_{0:T} = (z_0, z_1, \ldots, z_T)$ for each $x$. The generation process can be formulated as a finite-horizon Markov decision process (MDP): at each time step $t$, the "state" $s_t$ consists of the question $x$ together with the partial prefix $(z_0, \ldots, z_{t-1})$, and the "action" $a_t$ is the choice of next token $z_t$. To guide learning toward correct answers, the MDP is equipped with a reward function $r(s_t, a_t)$. It is typically defined as a binary outcome reward [16, 20] to determine whether the generated answer is correct.

According to existing RL-based fine-tuning paradigm [41, 43, 16], we are given a training dataset $\mathcal{D}_{\text{train}} = \{(x_i, y_i^*)\}_{i=1}^N$, where each $y_i^*$ is an oracle reasoning trace that leads to the correct answer. During fine-tuning, we use these traces both to calculate the reward and to guide learning. For each question $x_i$, we first generate candidate token sequences $z_{0:T} \sim \pi_\theta(\cdot \mid x_i)$. Then, we compute the reward $r(x_i, z_{0:T})$, which equals 1 if $z_{0:T}$ matches the oracle trace $y_i^*$. By maximizing the empirical sum of these rewards with the constraint of test-time budget $B_{\text{token}}$, we update $\pi_\theta$. The objective is:

$$\max_\theta \mathbb{E}_{x \sim \mathcal{D}_{\text{train}}} \mathbb{E}_{z_{0:T} \sim \pi_\theta(\cdot \mid x)} \big[ r(x, z_{0:T}) \big] \quad \text{s.t.} \ \mathbb{E}_{z_{0:T} \sim \pi_\theta(\cdot \mid x)} |z| \leq B_{token} \tag{1}$$

Through Eq.1, we train the LLM $\pi_\theta$ to capture both the need to produce correct answers (maximize the rewards) and the requirement of using limited token budgets (under a fixed compute budget).

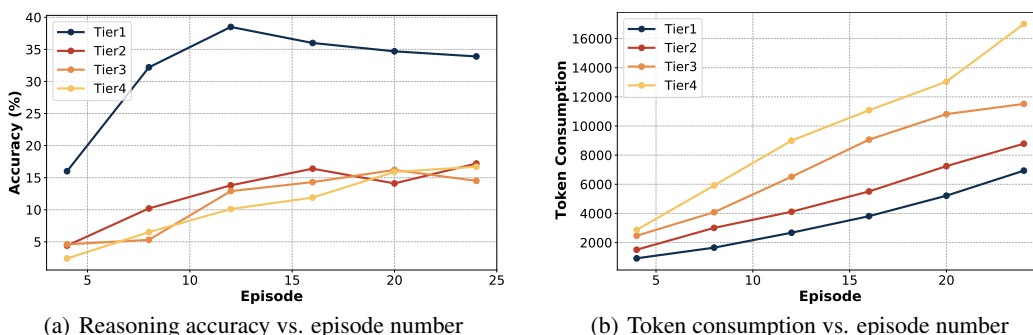

(a) Reasoning accuracy vs. episode number        (b) Token consumption vs. episode number

Figure 1: Results of DeepScaleR-1.5B-Preview across different tasks on Omni-MATH. We partition the generated reasoning chain into episodes, measuring accuracy $\mathrm{Acc}(k)$ and average token consumption $\overline{T}(k)$ at different episode depths. More details and results are shown in **Appendix G.1**.

## 3.2 Empirical Evidence

To ensure reasoning effectiveness, existing methods [43, 9, 53, 15, 49] mainly leverage test-time compute scaling to lengthen reasoning chains beyond what is minimally required for a correct answer, thereby more thoroughly exploring the solution space and improving final answer accuracy. These approaches typically use a sparse outcome reward for policy optimization, without feedback on intermediate reasoning steps [59, 55]. Using this framework, extending the chain carries no penalty, and even minimal accuracy gains from extra steps yield a positive signal. Thus, models resort to consuming additional tokens to secure the correctness of the final answer [31, 35]. However, under the fixed token budget $B_{token}$ considered in this paper, this extension may deplete the budget prematurely, undermining efficient reasoning in subsequent steps. To validate this argument, in this subsection, we conduct a series of experiments to analyze the token-utilization abilities of existing methods.

Specifically, in this experiment, we use the Omni-MATH benchmark [13] for evaluation, which comprises 4,428 questions across more than 33 domains, including algebra, discrete mathematics, geometry, number theory, etc. The questions are organized by human experts into four difficulty tiers (Tier 1-4) to assess model performance on tasks of varying complexity. We evaluate two base models [16], i.e., DeepScaleR-1.5B-Preview and DeepSeek-R1-Distill-Qwen-1.5B, running on the A100 GPU clusters with greedy decoding (temperature = 0) and a maximum generation length of 16,384 tokens. To probe different reasoning depths, we configure the prompt to wrap each logical step with '<think>'...'</think>' tags, segmenting the generated chain into up to $K = 30$ episodes. For each test question and each truncation point $k = 1, \ldots, 30$, we force-stop generation at the $k$-th '</think>' tag, append the prompt "Please output the final answer directly based on the above steps", and then use Omni-Judge to determine correctness. This yields the accuracy $\mathrm{Acc}(k)$ at episode depth $k$ and the average token consumption $\overline{T}(k)$ from the initial prompt to the truncation. Considering the impact of randomness, we introduce a $\mathrm{maj@4}(k)$ baseline: for each truncated context, we sample four continuations, take the majority-vote result as the final answer, and record its accuracy $\mathrm{maj@4}(k)$ and token cost. By plotting "accuracy vs. episode number" and "token consumption vs. episode number", we can visualize how performance evolves with reasoning depth across tasks of differing difficulty and compare sequential generation against majority-vote under the same compute budget.

From **Figures 1 and 6**, we can observe that: (i) Existing methods may fail to use test-time compute budgets efficiently, leading to wasted resources: both models have on average used more than twice the minimum tokens required. For example, $k = 16$ achieves accuracy comparable to or exceeding sequential generation at $k = 24$ with fewer tokens. (ii) The additional episodes add no new information and instead degrade performance due to context redundancy: for both models, $\mathrm{Acc}(k)$ peaks around $k \approx 16 - 20$ and then declines as $k$ increases. (iii) The questions of different difficulty tiers prefer different chain lengths: Tier 4 questions tend to benefit from longer chains, whereas Tier 1 questions can achieve correct results with short chains, where excessive reasoning depth may causes a marked accuracy drop (e.g., falls by over 5% at $k = 20$). These findings underscore the limitations of existing methods, which ignore the balance between reasoning effectiveness and efficiency.

### 3.3 Motivation Analysis

For obtaining a powerful LLM to address the above limitations under the settings in **Subsection 3.1**, in this subsection, we discuss the solutions that need to be incorporated to address the realistic challenges of existing methods. Based on these analyses, we design our framework (**Section 4**).

As illustrated in **Section 1** and **Subsection 3.2**, we discuss and demonstrate that, although outcome rewards can boost reasoning effectiveness by increasing the token budget, it sacrifices efficiency, i.e., many reasoning steps are invalid for the answer correctness; much longer CoTs are used than the correct answer really needed. To balance reasoning effectiveness and efficiency, we aim for an algorithm that yields positive gains at every reasoning step, enabling the model to achieve comparable reasoning performance within a constrained yet sufficient token budget. This requires augmenting the learning objective with dense, step-wise process rewards that immediately quantify each reasoning step's contribution to overall performance. By maximizing the cumulative reward across all reasoning steps, we encourage the model to generate tokens that most benefit the answer correctness. Therefore, the algorithm requires an effective dense process reward to ensure both effectiveness and efficiency: by maximizing total reward across all reasoning steps, we guarantee the reasoning effectiveness; by maximizing each token's contribution and preventing useless tokens, we ensure efficiency. Notably, some concurrent approaches attempt to reduce reasoning depth using task-specific process rewards [62, 67] and length penalties [57, 33]; however, they require manually designing high-quality process labels and task-specific evaluators, which is an expensive endeavor that may not generalize across tasks. Therefore, we must also address the second challenge of algorithmic generality.

Generality demands that the algorithm adapt to varied task requirements and remain effective. To achieve this, one would ideally define evaluation metrics that apply uniformly across all scenarios to construct reward functions. However, the heterogeneity of tasks makes it impractical to identify a single, fixed metric suitable for every case. Accordingly, we propose leveraging an internal model signal, e.g., the change in parameters, to quantify the contribution of intermediate reasoning steps. This measure directly reflects the amount of new knowledge the model acquires on the current task and is agnostic to input formats, task types, etc. Consequently, it enables sustained, reliable reward feedback without requiring additional annotations or retraining when new tasks are introduced.

In summary, the above analyses motivate us to design dense process rewards derived from internal model signals to jointly address both algorithmic efficiency and generality. By optimizing LLMs with these rewards, we consider both reasoning effectiveness and efficiency across different tasks.

## 4 Learning To Think

Based on the above analyses, we propose Learning to Think (L2T), an information-theoretic reinforcement fine-tuning framework for LLMs (with pseudo-code in **Appendix C**). The key is proposing a universal dense process reward for LLM optimization to adaptively allocate reasoning depth across tasks and prevent token waste. It recasts LLM optimization as an episodic learning problem to support episode-wise process reward calculation and optimizes LLM to maximize the contribution of tokens generated in each episode, ensuring reasoning effectiveness and efficiency. Specifically, in L2T, each query-response pair is segmented into successive reasoning episodes (**Subsection 4.1**), within which the model performs an adaptation update to increase the likelihood of a correct answer. To ensure the effectiveness of each adaptation and curb excessive reasoning, we then propose an information-theoretic dense process reward that immediately quantifies the progress of each episode with universal information gain to support policy optimization (**Subsection 4.2**). It can be efficiently estimated leveraging PAC-Bayes bounds and the Fisher information matrix with theoretical guarantees (**Theorem 4.2**). Finally, we optimize the LLMs by maximizing cumulative reward via RL (**Subsection 4.3**), ensuring both high reasoning effectiveness and efficiency across different tasks.

### 4.1 Problem Reformulation

In practice, reasoning questions vary widely, e.g., from simple arithmetic to complex mathematical proofs. Existing RL-based fine-tuning paradigm [51, 45] mainly treats the reasoning process of each question as a single episode and updates the policy via the final outcome reward. Under this setting, it is difficult to implement dense process rewards. To address this issue, we reformulate the problem mentioned in **Subsection 3.1** as an episodic RL problem. We treat each question as a task sampled

from a broader distribution and decompose the reasoning process into successive episodes. This decomposition allows us to assign process rewards at every intermediate episode, with each episode's reward reflecting its contribution to the answer correctness. Based on this dense reward, we can optimize the policy incrementally to maximize the performance gain of each episode.

Specifically, each question $x$ is viewed as a reasoning task $x \sim \mathcal{P}_{\text{task}}$. For a given $x$, the LLM carries out an internal inference procedure and emits a sequence of tokens $z_{0:T} = (z_0, \dots, z_T)$, as its final answer. To inject dense feedback, we insert '<think>...</think>' markers to break the full token stream into $K$ consecutive reasoning episodes. Each episode shares the accumulated context but serves as a natural checkpoint at which we can evaluate intermediate progress and assign episode-wise rewards. We treat the entire reasoning process as a length-$K$ MDP: at episode $k$, the state is $s_k = (x, z_{1:k-1})$; the action $z_k \sim \pi_\theta(\cdot \mid s_k)$ produces a token sequence $z_k = (z_k^1, \dots, z_k^{N_k})$. The reward consists of (i) a dense process reward $r_k^{\text{prg}}$ which measures the increase in the answer correctness probability after episode $k$, and (ii) a sparse outcome reward $r^{\text{out}} = r(x, z_{1:K}) \in \{0, 1\}$ that reflects the final answer's correctness (Eq.1). Note that we evaluate $r_k^{\text{prg}}$ at the episode level rather than per token to (i) reduce variance: individual tokens rarely determine final correctness and are easily corrupted by stopwords, sampling noise, etc.; (ii) lower cost: episode-level evaluation reduces expensive calls to the reward function. During fine-tuning, given the training set $\mathcal{D}_{\text{train}} = \{(x_i, y_i^*)\}_{i=1}^N$, we optimize the policy $\pi_\theta$ by maximizing the rewards across all the $x_i$, with objective:

$$\max_\theta \mathbb{E}_{x \sim \mathcal{D}_{\text{train}}} \mathbb{E}_{z_{1:K} \sim \pi_\theta(\cdot|s_k)} \left[ r^{\text{out}} + \sum_{k=1}^K r_k^{\text{prg}} \right], \quad \text{s.t. } \mathbb{E}_{z_{1:K} \sim \pi_\theta(\cdot|x)} |z| \le B_{token}, \quad (2)$$

where $B_{token}$ is the fixed test-time compute budget. Through Eq.2, the learned policy allocates its limited token budget ($B_{token}$) where it yields the greatest incremental benefit, expanding promising lines with higher process reward $r^{\text{prg}}$ and maximize overall task success $r^{\text{out}}$. In practice, however, the crux is the design of the newly introduced dense process reward $r^{\text{prg}}$, which need to satisfy following desiderata: (i) Relevance: faithfully measure the progress that episode $k$ contributes toward a correct solution; (ii) Efficiency: be cheap to compute; (iii) Generality: apply uniformly across tasks without bespoke engineering for each new domain. Recently proposed process-reward models [63, 36, 62, 67] predominantly focus on (i); however, they remain task-specific and depend on high-quality annotations, failing to satisfy (ii) and (iii). Therefore, designing such a dense, per-episode reward remains an open challenge, also the key to unlocking truly efficient reasoning in LLMs.

## 4.2 Learning Information-Theoretic Dense Process Reward

To address the above challenge, we propose a novel information-theoretic dense process reward. It is inspired by the information theory [3, 25] and consists of two constraints for reasoning effectiveness and efficiency: (i) a fitting information gain, which encourages the model to acquire key information about correctness in each episode; and (ii) a parameter compression penalty, which penalizes redundant information absorbed at each episode to maintain efficiency. This reward is agnostic to input format or task domain, which can be applied in various scenarios. In this subsection, we begin by explaining the meaning of the above two constraints with the proposed reward. Then, we provide the formal definition of this reward (**Definition 4.1**). Next, we explain how to efficiently compute it in practice: to handle the large parameter scale of LLMs, we develop an efficient approximation of this reward using PAC-Bayes theory and Fisher information matrix (**Theorem 4.2**). Finally, we illustrate why the proposed reward is effective, i.e., satisfying the three criteria mentioned in **Subsection 4.1**.

Firstly, we explain what the two components of the proposed reward are. Specifically, the fitting information gain measures the reduction in uncertainty about $Y$ given $X$ provided by parameters $\theta$ after each episode; formally, it is defined as the conditional mutual information $I(\theta; Y|X) = H(Y|X) - H(Y|X, \theta)$, where $H(Y|X) = -\sum_y p(y|X) \log p(y|X)$ denotes the uncertainty of $Y$ given $X$ alone and $H(Y|X, \theta)$ the residual uncertainty after observing $\theta$. The fitting gain for episode $k$, in which parameters update from $\theta_{k-1}$ to $\theta_k$, is $\Delta I_k = I(\theta_k; Y|X) - I(\theta_{k-1}; Y|X)$. Since direct computation is costly in LLMs, we approximate $\Delta I_k$ by the increase in the predicted correctness probability of the models, i.e., $\Delta I_k \approx J_r(\pi_\theta(\cdot|s_k, z_k)) - J_r(\pi_\theta(\cdot|s_k))$, which aligns with the direction of $\Delta I_k$ (**Appendix D.1**). It entails just two forward passes of $\pi_\theta$ without the need of gradient updates to estimate $\Delta I_k$, reducing computational overhead. The parameter compression penalty constrains redundant information captured from each episode, preventing excessive updates. It is defined as the mutual information between $\theta$ and the context $s_k$, denoted as $I(\theta; s_k) = \mathbb{E}_S[\text{KL}(p(\theta|s_k) \| p(\theta))]$. It quantifies the task-specific idiosyncrasies stored in $\theta$ [14], where larger mutual information implies

greater overfitting risk and unnecessary computational overhead. Thus, these terms align with our objective of evaluating reasoning effectiveness and efficiency in dense process reward design. More discussions, theoretical analyses, and the intuition behind are further illustrated in **Appendix D**.

Based on the above analyses, we then present the formal definition of the proposed reward.

**Definition 4.1** *Let the context before episode $k$ be $s_k = (x, z_{1:k-1})$, and the model generate the token sequence $z_k \sim \pi_\theta(\cdot \mid s_k)$. The dense process reward for episode $k$ can be expressed as:*

$$r_k^{\mathrm{prg}} = \underbrace{J_r\big(\pi_\theta(\cdot \mid s_k, z_k)\big) - J_r\big(\pi_\theta(\cdot \mid s_k)\big)}_{\textit{Fitting Information Gain}} - \beta \underbrace{\Big[I(\theta_k; s_k) - I(\theta_{k-1}; s_{k-1})\Big]}_{\textit{Parameter Compression Penalty}}. \tag{3}$$

*where $J_r(\cdot)$ denotes the correctness probability and $\beta > 0$ is a hyperparameter.*

Eq.3 indicates that the larger $r_k^{\mathrm{prg}}$, the update of this episode is more effective: the larger the first term, the greater improvement to predict the correct answer, increasing effectiveness; the smaller the second term, the less redundant information is absorbed, ensuring efficient and sufficient updates.

Obtaining **Definition 4.1**, we illustrate how to calculate it in practice. In the LLM setting, fitting information gain measures the contribution of each optimization episode to reasoning capability by tracking the change in the predicted correctness probability, i.e., the output distributions of $\pi_\theta$. In contrast, computing the parameter-compression penalty is more involved: it requires quantifying the mutual information increment between the parameters $\theta$ and the historical context $s_k$, where direct estimation in the large parameter space of LLMs is intractable. To address this, we introduce an efficient approximation that uses the low-rank parameter proxy $\tilde{\theta}$ with singular value decomposition (SVD) [6] and the Fisher information matrix [12, 37] to estimate the penalty term. We get:

**Theorem 4.2** *Given the low-rank parameter proxy $\tilde{\theta}_k$ and $\tilde{\theta}_{k-1}$ for parameters $\theta_k$ and $\theta_{k-1}$, assume that $\tilde{\theta}_k$ and $\tilde{\theta}_{k-1}$ follow Gaussian distribution, e.g., $p(\tilde{\theta}_k) = \mathcal{N}(\tilde{\theta}_k | \mu_k, \Sigma_k)$ where $\mu_k$ is the mean vector of the parameters and $\Sigma_k$ is the covariance matrix of the parameters, we get:*

$$I(\tilde{\theta}_k; s_k) - I(\tilde{\theta}_{k-1}; s_{k-1}) \simeq (\tilde{\theta}_k - \tilde{\theta}_{k-1})^\top \left( \nabla_{\tilde{\theta}_k} \log \pi_\theta(z_k | s_k) \nabla_{\tilde{\theta}_k} \log \pi_\theta(z_k | s_k)^\top \right) (\tilde{\theta}_k - \tilde{\theta}_{k-1}) \tag{4}$$

**Theorem 4.2** presents a method to estimate the intractable compression penalty (with proof in **Appendix B**). It simplifies the parameter space using SVD and assumes $\tilde{\theta}$ follows a Gaussian distribution based on [26], a common and mild assumption. Note that the low-rank approximation of $\theta \in \mathbb{R}^d$, i.e., obtaining $\tilde{\theta} \in \mathbb{R}^r$ ($r \ll d$), is to avoid direct computation of the Fisher matrix in high-dimensional space. We use SVD to extract the principal directions of variation in $\theta$ and retain the top $r$ components (with $r/d \approx 1\%$–10% for 1.5B models and $r/d \approx 0.1\%$–1% for 7B models), resulting in a low-rank surrogate $\tilde{\theta}$. Then, by computing the second derivative of the log-likelihood of $\theta$, we obtain the Fisher information matrix, which captures the effect of $\tilde{\theta}_k$ updates on the output and approximates the covariance calculation (**Lemma B.1**). The mutual information increment is then approximated using the second-order term of the Taylor expansion. This method significantly reduces the computational complexity (**Theorem D.2**) with limited approximation error (**Theorem D.3**) to support the computation of our proposed reward **Definition 4.1** in practice.

Finally, we explain why the proposed reward is effective. It satisfies the three criteria mentioned in **Subsection 4.1**: (i) for relevance: the fitting gain term measures how much episode $k$ improves the reasoning correctness, which is tightly aligned with task progress; (ii) for efficiency: it requires only one call for estimation per episode with **Theorem 4.2**, and the cost scales linearly with the number of episodes; (iii) for generality: neither term depends on task-specific models, just on the model's own correctness scores and parameter information gain, so the reward applies uniformly across tasks.

## 4.3 Optimizing LLM with Reinforcement Fine-Tuning

Based on the above-defined problem settings (**Subsection 4.1**) and proposed reward (**Subsection 4.2**), in this subsection, we introduce the optimization process of policy $\pi_\theta$ (i.e., the LLM).

Specifically, based on the reformulation in **Subsection 4.1**, under the RL framework relied upon by L2T, the optimization objective of the LLM can be decomposed into two parts: (i) for answer

correctness: maximizing the cumulative outcome reward $r^{\text{out}}$ obtained after a sequence of reasoning episodes to ensure the correctness of the answer; and (ii) for process progress: using the dense process reward to evaluate the improvement in correctness probability and the increase in model parameter information gain after each reasoning episode. This term is designed to capture the progress made at each episode of reasoning and optimize the model to maximize the incremental gain at each reasoning step. Among them, (i) corresponds to the standard fine-tuning objective in Eq.1, while (ii) depends on the process reward defined in **Subsection 4.2**. Therefore, the objective can be expressed as:

$$\max_{\theta} \mathbb{E}_{x \sim \mathcal{D}_{\text{train}}} \mathbb{E}_{z_{1:K} \sim \pi_{\theta}(\cdot|s_k)} \big[ r(x, z_{1:K}) + \alpha \textstyle\sum_{k=1}^{K} r_k^{\text{prg}} \big]$$
$$\text{s.t. } r_k^{\text{prg}} = J_r\big(\pi_{\theta}(\cdot \mid s_k, z_k)\big) - J_r\big(\pi_{\theta}(\cdot \mid s_k)\big) - \beta\Big[ I(\theta_k; s_k) - I(\theta_{k-1}; s_{k-1}) \Big], \tag{5}$$

where $\alpha$ is the importance weight (set to 1 for simplicity), and the $r_k^{\text{prg}}$ is calculated through **Theorem 4.2**. Through Eq.5, L2T optimizes the policy $\pi_{\theta}$ to achieve our goal of boosting effectiveness and efficiency in two parts: (i) The first part ensures the correctness of the final answer through the outcome reward $r(x, z_{0:K})$. (ii) The second part introduces the dense process reward $r_k^{\text{prg}}$, leveraging information-theoretic internal signals to assess the progress of each episode update. It encourages the model to maximize correctness at each step while avoiding redundant information accumulation. Thus, this optimization enables the model to efficiently utilize the limited token budget, progressively improving reasoning effectiveness, and ultimately achieving high-accuracy reasoning outputs.

**Practical Implementation with GRPO**    In practice, L2T is instantiated on top of GRPO to realize stable and efficient reinforcement optimization. Specifically, for each sampled question $x$, the old policy $\pi_{\theta_{\text{old}}}$ generates $N$ reasoning rollouts, where each is automatically split into consecutive episodes based on the prompt designed with '<think>'...'</think>' delimiters (**Appendix G.4** provides an exampele). This segmentation enables us to assign both the sparse outcome reward and the dense process reward at the episode level. Concretely, for episode $k$ in rollout $i$, the reward is defined as $R_{i,k} = \frac{1}{K_i} r_i^{\text{out}} + \alpha r_{i,k}^{\text{prg}}$, where the outcome reward is $r^{\text{out}} = \mathbb{1}[z_{1:K}$ leads to correct $y^*]$, and the dense process reward is $r_{i,k}^{\text{prg}} = \Delta I_k - \beta C_k$, with fitting information gain $\Delta I_k = J_r(\pi_{\theta}(\cdot \mid s_k, z_k)) - J_r(\pi_{\theta}(\cdot \mid s_k))$ and compression penalty $C_k = I(\theta_k; s_k) - I(\theta_{k-1}; s_{k-1})$ (see **Section 4.2**). Following GRPO, the episodic reward $R_{i,k}$ is further distributed to tokens using log-probability surprise as weights, i.e., $w_{i,t} \propto -\log p_{\theta_{\text{old}}}(z_{i,t} \mid s_{i,t})$, giving per-token rewards $r_{i,t}$. The truncated mean of these token-level rewards (95%) yields $\tilde{r}_i$, which is then normalized into a group-level advantage, i.e., $\hat{A}_i = \frac{\tilde{r}_i - \bar{\tilde{r}}}{\sigma_{\tilde{r}}}$. This group-level advantage is rescaled to tokens according to their relative contribution, i.e., $A_{i,t} = \hat{A}_i \cdot \frac{r_{i,t}}{\tilde{r}_i}$. Finally, the advantages $A_{i,t}$ are used in the clipped policy gradient objective of GRPO with an additional KL penalty to stabilize training. Through this implementation, L2T maintains the advantages of GRPO while extending it with explicit episodic decomposition and the integration of the proposed information-theoretic dense process reward, thereby achieving both effective and efficient reasoning optimization.

## 5  Experiments

In this section, we conduct extensive experiments on multiple reasoning benchmarks to verify the effectiveness and efficiency of L2T. More details and experiments are provided in **Appendix E-G**.

### 5.1  Experimental Settings

We evaluate on multiple reasoning benchmarks, including AIME24-25, AMC, MATH500 [18], MinervaMATH [27], and Omni-MATH [13] (see **Appendices E and G** for more benchmarks, e.g., code generation). We use DeepScaleR-1.5B-Preview and DeepSeek-R1-Distill-Qwen-1.5B as base models, which already generate reasoning traces marked with '<think>'. We compare L2T against (i) outcome reward-based RL methods (e.g., GRPO [41] for deepseek-model family, more in appendices) and (ii) test-time-compute-focused methods, e.g., length penalty [2] and process-reward approaches such as ReST-MCTS [62] and MRT [36]. Since DeepScaleR-1.5B-Preview has already undergone one round of fine-tuning on 40k math question-answer pairs, we fine-tune it on the 919 AIME questions (from 1989 to 2023); for DeepSeek-R1-Distill-Qwen-1.5B, we fine-tune on a random sample of 4,000 question-answer pairs from NuminaMath [28]. Both fine-tuning and evaluation use a maximum token budget of 16,384. For optimization, we set the learning rate to $1e^{-6}$, weight decay to 0.01, and batch

Table 1: Pass@1 performance on various math reasoning benchmarks. We compare base models trained with different fine-tuning approaches. The best results are highlighted in **bold**.

| Base model + Method | AIME 2024 | AIME 2025 | AMC 2023 | MATH500 | MinervaMATH | Avg. |
|---|---|---|---|---|---|---|
| **DeepScaleR-1.5B-Preview** | 42.8 | 36.7 | 83.0 | 85.2 | 24.6 | 54.5 |
| +outcome-reward RL (GRPO) | 44.5 (+1.7) | 39.3 (+2.6) | 81.5 (-1.5) | 84.9 (-0.3) | 24.7 (+0.1) | 55.0 (+0.5) |
| +length penalty | 40.3 (-2.5) | 30.3 (-6.4) | 77.3 (-5.7) | 83.2 (-2.0) | 23.0 (-1.6) | 50.8 (-3.7) |
| +ReST-MCTS | 45.5 (+2.7) | 39.5 (+2.8) | 83.4 (+0.4) | 84.8 (-0.4) | 23.9 (-0.7) | 55.4 (+0.9) |
| +MRT | 47.2 (+4.4) | 39.7 (+3.0) | 83.1 (+0.1) | 85.1 (-0.1) | 24.2 (-0.4) | 55.9 (+1.4) |
| +Ours | **48.5 (+5.7)** | **40.2 (+3.5)** | **85.4 (+2.4)** | **88.1 (+2.9)** | **26.5 (+1.9)** | **57.8 (+3.3)** |
| **DeepSeek-R1-Distill-Qwen-1.5B** | 28.7 | 26.0 | 69.9 | 80.1 | 19.8 | 44.9 |
| +outcome-reward RL (GRPO) | 29.8 (+1.1) | 27.3 (+1.3) | 70.5 (+0.6) | 80.3 (+0.2) | 22.1 (+2.3) | 46.0 (+1.1) |
| +length penalty | 27.5 (-1.2) | 22.6 (-3.4) | 64.4 (-5.5) | 77.1 (-3.0) | 18.8 (-1.0) | 42.0 (-2.9) |
| +ReST-MCTS | 30.5 (+1.8) | 28.6 (+2.6) | 72.1 (+1.2) | 80.4 (+0.3) | 20.3 (+0.5) | 46.4 (+1.5) |
| +MRT | 30.3 (+1.6) | 29.3 (+3.3) | 72.9 (+3.0) | 80.4 (+0.3) | 22.5 (+2.7) | 47.1 (+2.2) |
| +Ours | **32.9 (+4.2)** | **30.1 (+4.1)** | **73.5 (+3.6)** | **84.7 (+4.6)** | **24.5 (+4.7)** | **49.2 (+4.3)** |

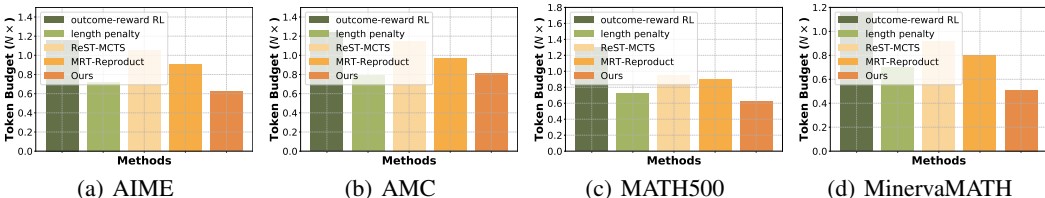

|                  (a) AIME                  |                  (b) AMC                  |                  (c) MATH500                  |                  (d) MinervaMATH                  |

Figure 2: Efficiency comparison across different benchmarks. We compute the token budget required for each benchmark and treat the budget of the base model w/o fine-tuning as reference ($1\times$).

size to 256. To approximate the parameter-compression penalty, we employ a one-layer MLP with a Fisher information-matrix damping coefficient of $1 \times 10^{-5}$. The hyperparameters $\alpha$ (Eq.5) and $\beta$ (Eq.3) are set to 0.8 and 0.6, respectively. All experiments are run on the A100 GPU clusters. More details of implementation and hyperparameters are provided in **Appendix F**.

## 5.2 Effectiveness and Efficiency Analysis

**Achieve better reasoning with higher efficiency** We compare L2T with the baselines across all the benchmarks and base models, recording both pass@1 accuracy and the required token budget. To reduce variance from limited samples, we use 20 outputs per question. **Table 1** and **Figure 2** show that L2T attains SOTA performance, achieving the highest reasoning effectiveness with the smallest token budget. For example, compared to outcome-reward-based methods, L2T delivers over a 3.7% gain in pass@1 and roughly doubles token efficiency; compared to methods focused on test-time compute, it achieves more than a 2% accuracy improvement while reducing the token budget by 20%. Moreover, L2T consistently outperforms baselines on multiple datasets with distributions different from the training data, further demonstrating its effectiveness across diverse tasks. Besides, we also assess the performance of L2T on more base models with different scales, e.g., DeepSeek-R1-Distill-Qwen-7B and Qwen2-7B-Instruct, and more reasoning tasks, e.g., code generation tasks. Notably, mathematical reasoning and code generation serve as classic benchmarks for testing an LLM's complex reasoning ability [43, 16, 32]. The results in **Appendix G.2**, demonstrate the advantage of L2T. These results confirm the superiority of our approach, which achieves effective reasoning with higher efficiency.

**More efficient use of test-time compute** Based on **Subsection 5.1**, we sample reasoning trajectories across various benchmarks with a fixed token context window. We truncate these trajectories at different token budgets and evaluate performance. **Figure 3** shows the success rate against token consumption. We observe that (i) under the same token budget, L2T achieves higher reasoning accuracy; (ii) L2T consumes only 18% of the tokens required by the base model, 50% of those used by outcome-reward fine-tuning, and approximately 20% fewer tokens than process-reward models. These results demonstrate that L2T more effectively leverages test-time compute.

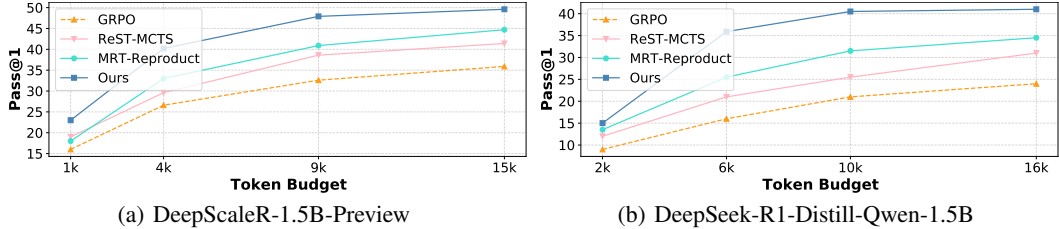

(a) DeepScaleR-1.5B-Preview  (b) DeepSeek-R1-Distill-Qwen-1.5B

Figure 3: Pass@1 vs. token budget of different methods on AIME. We record the model reasoning accuracy under different maximum token budgets to evaluate the ability of using test-time compute.

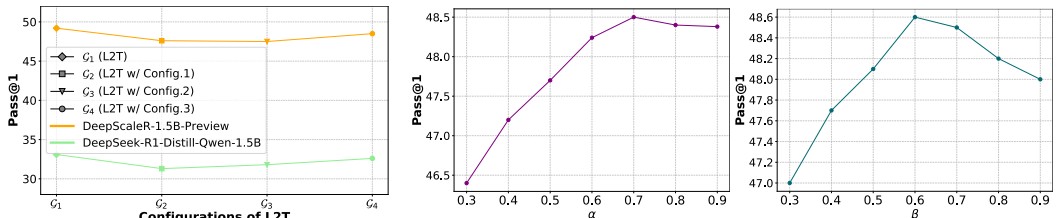

Figure 4: Effect of L2T components.  Figure 5: Parameter sensitivity of $\alpha$ and $\beta$

## 5.3 Ablation study

We conduct a series of ablation studies to evaluate the contribution of each component within L2T, the best parameterization and implementation choices, etc. See **Appendix G** for more results.

**The effect of different components.** We evaluate three alternative configurations: (i) replacing the fitting information gain with a task-specific reward model; (ii) removing the parameter-compression penalty; and (iii) substituting the low-rank approximation with random sampling of 30% layers. Notably, the overall contribution of our reward has already been demonstrated in **Subsection 5.2**. From **Figure 4** and **Appendix G.3**, we observe that both the fitting information gain and the parameter-compression penalty are critical for LLM reasoning; although random sampling is faster than low-rank approximation, it introduces additional error. These findings underscore the advantages of our design.

**Parameter sensitivity.** We determine the hyperparameters of L2T by evaluating reasoning performance across benchmarks. Both $\alpha$ and $\beta$ are swept over $[0.3, 0.9]$. We first use grid search to screen the parameters with a difference of 0.05, then refine it with 0.01, recording the average outcome. As shown in **Figure 5**, the optimal setting is $\alpha = 0.8$ and $\beta = 0.6$, also our choices.

## 6  Conclusion

In this paper, we propose Learning to Think (L2T), an information-theoretic reinforcement fine-tuning framework for LLMs. It reformulates LLM optimization as an episodic RL problem and proposes a universal information-theoretic dense process reward to support policy optimization, i.e., incentivizing the model to focus on progress in each episode, thus achieving great reasoning performance under a minimal token budget. Extensive experiments on multiple complex reasoning benchmarks demonstrate the advantages of L2T in both reasoning effectiveness and efficiency.

## Acknowledgements

The authors would like to thank the anonymous reviewers for their valuable comments. This work is supported by the National Natural Science Foundation of China (No. 62506355).

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

# Appendix

The appendix is organized as follows:

## A   List of Notations

We list the definitions of all notations from the main text as follows:

☐ **Symbols of Problem Settings**

- $\mathcal{D}_{\text{train}} = \{(x_i, y_i^*)\}_{i=1}^N$: training dataset.
- $x_i$: $i$-th question.
- $y_i^*$: the oracle reasoning trace that leads to the correct answer.
- $\pi_\theta$: the LLM (treated as a stochastic policy).
- $z_{0:T} \sim \pi_\theta(\cdot \mid x_i)$: the output sequence for $x_i$.
- $B_{\text{token}}$: the token budget.
- $r(x_i, z_{0:T})$: the reward, which equals 1 if $z_{0:T}$ matches the oracle trace $y_i^*$.

☐ **Symbols of Problem Reformulation**

- $\mathcal{D}_{\text{train}} = \{(x_i, y_i^*)\}_{i=1}^N$: training dataset.
- $\pi_\theta$: the LLM (treated as a stochastic policy).
- $z_{0:T} \sim \pi_\theta(\cdot \mid x_i)$: the output sequence for $x_i$.
- $B_{\text{token}}$: the token budget.
- $K$: the number of episodes.
- $s_k = (x, z_{1:k-1})$: the state of $k$-th episode.
- $z_k \sim \pi_\theta(\cdot \mid s_k)$: the token sequence $z_k = (z_k^1, \ldots, z_k^{N_k})$ for the $k$-th episode of $x_i$.
- $N_k$: the number of tokens in $z_k$.
- $r^{\text{out}} = r(x, z_{1:K}) \in \{0, 1\}$: the outcome reward indicating final correctness.
- $r_k^{\text{prg}}$: the dense process reward at episode $k$ (Eq.3).
- $J_r(\cdot)$: the predicted correctness probability.

☐ **Information-Theoretic Quantities**

- $H(Y|X)$: conditional entropy of $Y$ given $X$ (uncertainty without parameters).
- $H(Y|X, \theta)$: residual uncertainty of $Y$ given $X$ and parameters $\theta$.
- $I(\theta; Y|X)$: conditional mutual information between $\theta$ and $Y$ given $X$.
- $\Delta I_k$: fitting information gain contributed by episode $k$.
- $I(\theta; s_k)$: mutual information between parameters and context.

☐ **Parameters, Proxies, and Updates**

- $\theta$, $\theta_{k-1}$, $\theta_k$: model parameters before/after episode $k$.
- $\tilde{\theta}$, $\tilde{\theta}_{k-1}$, $\tilde{\theta}_k$: low-rank proxies of $\theta$ via SVD (retain top $r$ components).
- $p(\tilde{\theta}_k) = \mathcal{N}(\tilde{\theta}_k \mid \mu_k, \Sigma_k)$: Gaussian assumption for the low-rank proxy.
- $r/d$: retained-rank ratio (e.g., $1\% \sim 10\%$ for 1.5B; $0.1\% \sim 1\%$ for 7B).

☐ **GRPO Integration and Token-Level Credit**

- $R_{i,k} = \frac{1}{K_i} r_i^{\text{out}} + \alpha \, r_{i,k}^{\text{prg}}$: per-episode reward for rollout $i$ (with process-weight $\alpha$).

- $w_{i,t} \propto -\log p_{\theta_{\text{old}}}(z_{i,t} \mid s_{i,t})$: token weights by log-probability surprise.
- $r_{i,t}$: token-level rewards obtained by distributing $R_{i,k}$ using $w_{i,t}$.
- $\tilde{r}_i$: 95% truncated mean of token rewards for rollout $i$.
- $\hat{A}_i = (\tilde{r}_i - \bar{\tilde{r}})/\sigma_{\tilde{r}}$: group-level advantage (standardized).
- $A_{i,t} = \hat{A}_i \cdot \dfrac{r_{i,t}}{\tilde{r}_i}$: token-level advantage rescaled by relative contribution.

# B  Proofs

In this section, we provide the proofs of **Theorem 4.2** in the main text. The theoretical analyses of the proposed dense process reward are provided in **Appendix D.1**.

*Proof.*  Our proof comprises three parts: (i) Part I, Information-gain derivation: Beginning with the fundamental definition of mutual information and assuming a Gaussian form, we derive the information gain $\Delta I = I(\tilde{\theta}_k; s_k) - I(\tilde{\theta}_{k-1}; s_{k-1})$, which under the Gaussian assumption admits the approximation and get $\Delta I \approx (\tilde{\theta}_k - \tilde{\theta}_{k-1})^\top \Sigma_0^{-1}(\tilde{\theta}_k - \tilde{\theta}_{k-1})$. (ii) Part II, Laplace approximation for variance estimation: We then employ the Laplace approximation to estimate the intractable posterior covariance in $\Delta I$ by way of the Fisher information matrix. (iii) Part III, Derivation of the compression penalty: Finally, we use the Fisher information matrix to derive the parameter-compression penalty that regularizes redundant information accumulation.

**Part I**  We begin by giving the concept of the mutual information in which the information is stored in the weights. It can be expressed as:

$$I(\tilde{\theta}; s) = \mathbb{E}_s\left[\mu_{KL}\big(p(\tilde{\theta} \mid s) \parallel p(\tilde{\theta})\big)\right]. \tag{6}$$

Assume that both the prior and the posterior are Gaussian distributions, we get:

$$p(\tilde{\theta}) = \mathcal{N}\big(\tilde{\theta} \mid \tilde{\theta}_0, \Sigma_0\big), \quad p(\tilde{\theta} \mid s) = \mathcal{N}\big(\tilde{\theta} \mid \tilde{\theta}_s, \Sigma_s\big), \tag{7}$$

then the mutual information in which the information is stored in the weights becomes:

$$I(\tilde{\theta}; s) = \mu_{KL}\big(p(\tilde{\theta} \mid s) \parallel p(\tilde{\theta})\big) = \tfrac{1}{2}\left[\ln \frac{\det \Sigma_s}{\det \Sigma_0} - d + (\tilde{\theta}_s - \tilde{\theta}_0)^\top \Sigma_0^{-1}(\tilde{\theta}_s - \tilde{\theta}_0) + \text{tr}\big(\Sigma_0^{-1}\Sigma_s\big)\right], \tag{8}$$

where $\det(\cdot)$ and $\text{tr}(\cdot)$ are the determinant and trace, $d$ is the dimension of parameter $\theta$ and is a constant for a specific NN architecture. If we assume $\Sigma_s \approx \Sigma_0$ following [3], then the logarithmic determinant and the trace term are constants, and the mutual information simplifies to:

$$I(\tilde{\theta}; s) \approx \tfrac{1}{N} \mathbb{E}_s\left[(\tilde{\theta}_s - \tilde{\theta}_0)^\top \Sigma_0^{-1}(\tilde{\theta}_s - \tilde{\theta}_0)\right], \tag{9}$$

where $N$ represents the number of trajectories for LLM optimization.

*Why a Gaussian distribution is a common and mild assumption under this context.*  During LLM training, parameter updates $\theta$ can be viewed as the accumulation of numerous small stochastic gradient steps. Each step introduces a small, random perturbation in parameter space, effectively acting as the sum of many independent random variables. When these perturbations are sufficiently numerous and diverse, the overall distribution of parameter changes tends toward a Gaussian distribution, as implied by the Central Limit Theorem. Therefore, when estimating the compression term, we model the low-rank surrogate $\tilde{\theta}$ (obtained via SVD) using a Gaussian distribution. This assumption aligns with covariance approximation techniques in the PAC-Bayes framework and Fisher information-based methods, allowing us to derive a closed-form expression for mutual information and significantly reduce computational cost.

Next, recalling our proposed parameter compression penalty, we care about the increase in mutual information between two adjacent proxy parameters, that is:

$$\Delta I = I(\tilde{\theta}_k; s_k) - I(\tilde{\theta}_{k-1}; s_{k-1}). \tag{10}$$

Consider that the agents at step $k$ and step $(k-1)$ be centered on the same prior $\tilde{\theta}_0$, then we get:

$$I(\tilde{\theta}_k; s_k) \approx \tfrac{1}{N}(\tilde{\theta}_k - \tilde{\theta}_0)^\top \Sigma_0^{-1}(\tilde{\theta}_k - \tilde{\theta}_0), \tag{11}$$

and

$$I(\tilde{\theta}_{k-1}; s_{k-1}) \approx \tfrac{1}{N}\,(\tilde{\theta}_{k-1} - \tilde{\theta}_0)^\top \Sigma_0^{-1}(\tilde{\theta}_{k-1} - \tilde{\theta}_0). \tag{12}$$

Based on the above formula, we can directly subtract and obtain:

$$\Delta I \approx \frac{1}{N}\Big[(\tilde{\theta}_k - \tilde{\theta}_0)^\top \Sigma_0^{-1}(\tilde{\theta}_k - \tilde{\theta}_0) - (\tilde{\theta}_{k-1} - \tilde{\theta}_0)^\top \Sigma_0^{-1}(\tilde{\theta}_{k-1} - \tilde{\theta}_0)\Big]. \tag{13}$$

Using the quadratic increment identity, i.e., $x^\top A x - y^\top A y = (x - y)^\top A(x + y)$,, and for small step updates $\tilde{\theta}_k - \tilde{\theta}_{k-1}$, discarding the constant factor, we have $\Delta I \approx (\tilde{\theta}_k - \tilde{\theta}_{k-1})^\top \Sigma_0^{-1}(\tilde{\theta}_k - \tilde{\theta}_{k-1})$.

**Part II**  However, it is difficult to calculate the covariance matrix for $\Delta I$. To address this, we use the Laplace approximation $\Sigma_0^{-1} \approx F_{\hat{\theta}}$ and the matrix representation of Fisher information. We get:

**Lemma B.1**  *Under the Gaussian-constant covariance assumption, the prior covariance matrix $\Sigma_0$ can be approximated by the inverse of the Fisher information matrix:*

$$\Sigma_0 \approx F_{\hat{\theta}}^{-1}, \text{ s.t. } F_{\hat{\theta}} = \mathbb{E}_{z \sim \pi_\theta(\cdot | s_k)}\Big[\nabla_{\tilde{\theta}_k} \log \pi_\theta(z \mid s_k)\,\nabla_{\tilde{\theta}_k} \log \pi_\theta(z \mid s_k)^\top\Big]. \tag{14}$$

*Proof.*  For a given state $s_k$, the posterior distribution $p(\tilde{\theta}_k \mid s_k) \propto p(s_k \mid \tilde{\theta}_k)\,p(\tilde{\theta}_k)$, take the logarithm to get the unstandardized posterior $L(\tilde{\theta}_k) = \log p(s_k \mid \tilde{\theta}_k) + \log p(\tilde{\theta}_k)$. Among them,

$$\log p(s_k \mid \tilde{\theta}_k) = \sum_{i=1}^{N} \log \pi_\theta(z_i \mid s_k), \quad \log p(\tilde{\theta}_k) = -\tfrac{1}{2}\,(\tilde{\theta}_k - \tilde{\theta}_0)^\top \Sigma_p^{-1}(\tilde{\theta}_k - \tilde{\theta}_0). \tag{15}$$

Take the partial derivative of the $m$th component of $\tilde{\theta}_k$, we get:

$$\frac{\partial}{\partial \tilde{\theta}_{k,m}} \log p(s_k \mid \tilde{\theta}_k) = \sum_{i=1}^{N} \frac{\partial}{\partial \tilde{\theta}_{k,m}} \log \pi_\theta(z_i \mid s_k) \equiv \sum_{i=1}^{N} g_{i,m}, \tag{16}$$

where $g_{i,m} = \partial_m \log \pi_\theta(z_i \mid s_k)$. Continue to take the partial derivative of the $n$th component. Taking the partial derivative of the component, we get:

$$\frac{\partial^2}{\partial \tilde{\theta}_{k,n}\,\partial \tilde{\theta}_{k,m}} \log p(s_k \mid \tilde{\theta}_k) = \sum_{i=1}^{N} \frac{\partial}{\partial \tilde{\theta}_{k,n}}\big[g_{i,m}\big] = \sum_{i=1}^{N} \underbrace{\frac{\partial^2}{\partial \tilde{\theta}_{k,n}\,\partial \tilde{\theta}_{k,m}} \log \pi_\theta(z_i \mid s_k)}_{h_{i,mn}}, \tag{17}$$

Therefore, the Hessian matrix of the log-likelihood is:

$$H(\tilde{\theta}_k) = \nabla_{\tilde{\theta}_k}^2 \mathcal{L}(\tilde{\theta}_k) = \nabla_{\tilde{\theta}_k}^2 \Big[\underbrace{\sum_{i=1}^{N} \log \pi_\theta(z_i \mid s_k)}_{\log p(s_k \mid \tilde{\theta}_k)} + \underbrace{\Big(-\tfrac{1}{2}(\tilde{\theta}_k - \tilde{\theta}_0)^\top \Sigma_p^{-1}(\tilde{\theta}_k - \tilde{\theta}_0)\Big)}_{\log p(\tilde{\theta}_k)}\Big]$$

$$= \underbrace{\sum_{i=1}^{N} \nabla_{\tilde{\theta}_k}^2 \log \pi_\theta(z_i \mid s_k)}_{H_{\text{lik}}} + \underbrace{\nabla_{\tilde{\theta}_k}^2 \Big[-\tfrac{1}{2}(\tilde{\theta}_k - \tilde{\theta}_0)^\top \Sigma_p^{-1}(\tilde{\theta}_k - \tilde{\theta}_0)\Big]}_{H_{\text{prior}}} \tag{18}$$

$$= \sum_{i=1}^{N} \Big[\frac{\partial^2}{\partial \tilde{\theta}_{k,m}\,\partial \tilde{\theta}_{k,n}} \log \pi_\theta(z_i \mid s_k)\Big]_{m,n=1}^{d} + \Sigma_p^{-1}$$

Let $\hat{\theta} = \arg\max_{\tilde{\theta}} L(\tilde{\theta})$ be the MAP (maximum a posteriori) estimate point of the posterior, and perform a second-order Taylor expansion of $L(\tilde{\theta})$ at $\hat{\theta}$ to obtain:

$$\mathcal{L}(\tilde{\theta}_k) \approx \mathcal{L}(\hat{\theta}_k) + \tfrac{1}{2}(\tilde{\theta}_k - \hat{\theta}_k)^\top H(\hat{\theta}_k)\,(\tilde{\theta}_k - \hat{\theta}_k). \tag{19}$$

then the Gaussian approximation is obtained as $p(\tilde{\theta}_k \mid s_k) \approx \mathcal{N}(\tilde{\theta}_k \mid \hat{\theta}_k,\ (-H(\hat{\theta}_k))^{-1})$ and the prior covariance should be $\Sigma_0 = \text{Cov}\big[p(\tilde{\theta}_k \mid s_k)\big] \approx (-H(\hat{\theta}_k))^{-1}$.

Next, we discuss the relationship between Hessian and Fisher information. First, the observed Fisher of log-likelihood is defined as $-\sum_i \nabla^2_{\tilde{\theta}_k} \log \pi_\theta(z_i|s_k) = \sum_i \nabla_{\tilde{\theta}_k} \log \pi_\theta(z_i|s_k) \nabla_{\tilde{\theta}_k} \log \pi_\theta(z_i|s_k)^\top$. When the third-order and higher derivatives are ignored, it can be approximated by:

$$-H_{\mathrm{lik}}(\hat{\theta}_k) \approx \sum_{i=1}^{n_k} \nabla_{\tilde{\theta}_k} \log \pi_\theta(z_i \mid s_k) \nabla_{\tilde{\theta}_k} \log \pi_\theta(z_i \mid s_k)^\top. \tag{20}$$

Further take the expectation of the policy distribution and define:

$$F_{\hat{\theta}_k} = \mathbb{E}_{z \sim \pi_\theta(\cdot|s_k)} \left[ \nabla_{\tilde{\theta}_k} \log \pi_\theta(z \mid s_k) \nabla_{\tilde{\theta}_k} \log \pi_\theta(z \mid s_k)^\top \right]. \tag{21}$$

For a flat prior or an oracle prior of $\Sigma_p \gg H_{\mathrm{lik}}^{-1}$, we have $\|H_{\mathrm{prior}}\| \ll \|H_{\mathrm{lik}}\|$, so the effect of $H_{\mathrm{prior}}$ on the population Hessian can be ignored in the large sample limit. Combining the above, we get:

$$-H(\hat{\theta}_k) = -\left(H_{\mathrm{lik}} + H_{\mathrm{prior}}\right) \approx F_{\hat{\theta}_k}, \quad \Sigma_0 \approx (-H(\hat{\theta}_k))^{-1} \approx F_{\hat{\theta}_k}^{-1}. \tag{22}$$

Thus, we complete the proof of the **Lemma B.1**.

**Part III**  Bring the above results back to **Theorem 4.2**, $\Delta I \approx (\tilde{\theta}_k - \tilde{\theta}_{k-1})^\top \Sigma_0^{-1} (\tilde{\theta}_k - \tilde{\theta}_{k-1})$ can be derived as:

$$\Delta I \simeq (\tilde{\theta}_k - \tilde{\theta}_{k-1})^\top \left( \nabla_{\tilde{\theta}_k} \log \pi_\theta(z_k \mid s_k) \nabla_{\tilde{\theta}_k} \log \pi_\theta(z_k \mid s_k)^\top \right) (\tilde{\theta}_k - \tilde{\theta}_{k-1}), \tag{23}$$

Thus, we get:

$$I(\tilde{\theta}_k; s_k) - I(\tilde{\theta}_{k-1}; s_{k-1}) \simeq (\tilde{\theta}_k - \tilde{\theta}_{k-1})^\top \left( \nabla_{\tilde{\theta}_k} \log \pi_\theta(z_k|s_k) \nabla_{\tilde{\theta}_k} \log \pi_\theta(z_k|s_k)^\top \right) (\tilde{\theta}_k - \tilde{\theta}_{k-1}) \tag{24}$$

We complete the proof of **Theorem 4.2**.

## C  Pseudo-Code

The pseudo-code of L2T is shown in **Algorithm 1**, providing the main steps of L2T with GRPO.

---
**Algorithm 1** Pseudo-Code of L2T (GRPO Version)
---
**Require:** Initial policy $\pi_\theta$; prompt distribution $\mathcal{D}$; hyperparameters $\alpha$ and $\beta$
1: **for** step $= 1$ **to** $N$ **do**
2:     Sample a batch $D_b$ from $\mathcal{D}$
3:     Set old policy $\pi_{\theta_{\mathrm{old}}} \leftarrow \pi_\theta$
4:     **for** each query $x_i \in D_b$ **do**
5:         Sample $N$ rollouts $\{y_0, y_1, \cdots, y_{N-1}\} \sim \pi_{\theta_{\mathrm{old}}}(\cdot \mid x_i)$
6:         **for** each rollout $y_i$ **do**
7:             Compute $r_{i,k}^{\mathrm{prg}}$ via Definition 4.1 and Theorem 4.2 for each episode $k$
8:             Compute $r_i^{\mathrm{out}}$ following [16]
9:             Compute per-episode reward $R_{i,k} = \frac{1}{K_i} r_i^{\mathrm{out}} + \alpha\, r_{i,k}^{\mathrm{prg}}$
10:            Set surprise weights $w_{i,t} \propto -\log p_{\theta_{\mathrm{old}}}(z_{i,t} \mid s_{i,t})$
11:            Assign per-token rewards $r_{i,t} \leftarrow \frac{w_{i,t}}{\sum_{t' \in k} w_{i,t'}} \cdot R_{i,k}$
12:         **end for**
13:         Compute truncated mean $\tilde{r}_i \leftarrow \mathrm{TruncMean}_{95\%}(\{r_{i,t}\}_t)$
14:         Compute group-level advantage $\hat{A}_i = \frac{\tilde{r}_i - \bar{\tilde{r}}}{\sigma_{\tilde{r}}}$
15:         Rescale to token-level advantages $A_{i,t} = \hat{A}_i \cdot \frac{r_{i,t}}{\tilde{r}_i}$
16:     **end for**
17:     Update $\pi_\theta$ via the GRPO objective
18: **end for**
19: **return** $\pi_\theta$
---

# D  More Discussion

## D.1  Theoretical Analyses about the Proposed Reward

In this subsection, we present the theoretical analysis for the proposed information-theoretic dense process reward. Specifically, we first explain why the increment in model prediction accuracy can be used to approximate $\Delta I_k$ (**Proposition D.1**). Next, we provide a theoretical analysis demonstrating that, under the current approximation, the computation method in **Theorem 4.2** significantly reduces computational complexity (**Theorem D.2**), while the approximation error (**Theorem D.3**) remains bounded, thereby supporting the practical computation of the proposed reward.

As mentioned in **Subsection 4.2**, the fitting information gain quantifies the reduction in uncertainty about $Y$ provided by the model parameters $\theta$ after each episode. Formally, it is defined as the conditional mutual information $I(\theta; Y \mid X) = H(Y \mid X) - H(Y \mid X, \theta)$, where $H(Y \mid X) = -\sum_y p(y \mid X) \log p(y \mid X)$ represents the uncertainty of $Y$ given $X$ alone, and $H(Y \mid X, \theta)$ is the residual uncertainty after observing $\theta$. The fitting gain for episode $k$, during which the parameters update from $\theta_{k-1}$ to $\theta_k$, is given by $\Delta I_k = I(\theta_k; Y \mid X) - I(\theta_{k-1}; Y \mid X)$. Given the computational expense of directly calculating mutual information in large models, we approximate $\Delta I_k$ by the increase in the model's predicted correctness probability. Specifically, we use the approximation $\Delta I_k \approx J_r(\pi_\theta(\cdot \mid s_k, z_k)) - J_r(\pi_\theta(\cdot \mid s_k))$, which captures the direction of $\Delta I_k$ and requires only two evaluations of the distribution per episode. Then, we get:

**Proposition D.1** *Given an episode $k$, where the model parameters are updated from $\theta_{k-1}$ to $\theta_k$, the fit gain for this episode is defined as $\Delta I_k = I(\theta_k; Y \mid X) - I(\theta_{k-1}; Y \mid X)$, where $I(\theta; Y \mid X)$ represents the mutual information between the model parameters $\theta$ and the labels $Y$, conditioned on the input $X$. We have:*

$$\Delta I_k \approx J_r \left( \pi_\theta(\cdot \mid s_k, z_k) \right) - J_r \left( \pi_\theta(\cdot \mid s_k) \right), \tag{25}$$

*where $J_r(\pi_\theta)$ denotes the reward function under the policy $\pi_\theta$, and $\pi_\theta(\cdot \mid s_k, z_k)$ and $\pi_\theta(\cdot \mid s_k)$ represent the updated policy and the policy prior to the update, respectively. This approximation aligns with the direction of the mutual information increment.*

*Proof.* We start by expressing the mutual information $I(\theta; Y \mid X)$ between model parameters $\theta$ and output labels $Y$ conditioned on input $X$. This is formally defined as:

$$
\begin{aligned}
I(\theta; Y \mid X) &= H(Y \mid X) - H(Y \mid X, \theta) \\
&= -\sum_y p(y \mid X) \log p(y \mid X) + \sum_y p(y \mid X, \theta) \log p(y \mid X, \theta) \\
&= \sum_y p(y \mid X) \log \frac{p(y \mid X)}{p(y \mid X, \theta)}
\end{aligned}
\tag{26}
$$

where $H(Y \mid X)$ is the entropy of $Y$ given $X$ (uncertainty of the labels given the input), $H(Y \mid X, \theta)$ is the conditional entropy of $Y$ given both $X$ and $\theta$, $p(y \mid X)$ is the conditional probability distribution of label $Y$ given input $X$, and $p(y \mid X, \theta)$ is the conditional probability distribution of label $Y$ given input $X$ and model parameters $\theta$.

The change in mutual information between two episodes (from $\theta_{k-1}$ to $\theta_k$) is given by $\Delta I_k = I(\theta_k; Y \mid X) - I(\theta_{k-1}; Y \mid X)$. This represents the gain in the model's ability to predict $Y$ given $X$, as the parameters are updated from $\theta_{k-1}$ to $\theta_k$.

In RL-based LLM optimization, the reward function $J_r(\pi_\theta)$ can be interpreted as the expected accuracy of the model, which evaluates how well the model's predictions align with the correct answer. For a given policy $\pi_\theta$, the reward function is defined as $J_r(\pi_\theta) = \mathbb{E}_{z \sim \pi_\theta(\cdot \mid s_k)}[r(z)]$ where $z$ denotes the model's output and $r(\cdot)$ measures how correctly the model generate the answer. In this case, the difference between the model's output probability (i.e., $p(y \mid X, \theta)$) and the label $Y$ directly affects the model's reasoning accuracy.

From the perspective of information theory [3, 25], the model's fitting information gain $\Delta I$ reflects the change in the difference between the model's inference answer and the standard answer (label) before and after the parameter update. High mutual information means that the relationship between the model's inference and the standard answer is stronger, in other words, given $X$ and $\theta$, the model

is able to predict $Y$ more accurately. The reward function also reflects this by measuring the accuracy of the model in its predictions. In other words, increasing mutual information actually improves the accuracy of the model, and accuracy can be quantified by the reward function. Therefore, the reward function can reflect the information gain obtained by the model in label prediction, or in other words, the reward function can reflect the improvement in the accuracy of the model's predictions. The accuracy improvement and mutual information increment have similar directions: both reflect the improvement in the ability to capture label information. Recall the problem settings, the model updates the policy $\pi_\theta$ through the parameter $\theta$, which affects the prediction accuracy of the model. The updated policy $\pi_\theta(\cdot \mid s_k, z_k)$ and the pre-update policy $\pi_\theta(\cdot \mid s_k)$ will lead to changes in prediction accuracy, thereby affecting the value of the reward function. Since accuracy is related to mutual information, we can approximate the mutual information increment by the difference in reward function. Therefore, we have:

$$\Delta I_k \approx J_r \left( \pi_\theta(\cdot \mid s_k, z_k) \right) - J_r \left( \pi_\theta(\cdot \mid s_k) \right) \tag{27}$$

This shows that the difference in reward function is consistent with the direction of the increase in mutual information, both reflecting the increase in the amount of information when the model predicts the label $Y$ (correct answer).

In the context of LLMs, the fitting information gain quantifies the contribution of each optimization episode to the reasoning ability by tracking changes in the predicted correctness probability, i.e., the output distributions of $\pi_\theta$. In contrast, calculating the parameter-compression penalty is more complex: it involves estimating the mutual information increment between the model parameters $\theta$ and the historical context $s_k$. Direct computation of this increment is intractable in the large parameter space of LLMs. To overcome this challenge, we propose an efficient approximation in **Theorem 4.2** to estimate the penalty term. Next, we demonstrate that this method significantly reduces the computational complexity (**Theorem D.2**) with limited approximation error (**Theorem D.3**) to support the computation of our proposed reward.

First, we prove that in **Theorem 4.2**, computing the parameter-compression penalty via low-rank approximation and Fisher matrix estimation achieves speedups of several orders of magnitude.

**Theorem D.2** *Let the parameter dimension be $d$ and the rank cutoff value be $r$ ($r \ll d$). Compared to the original full non-approximate computation, estimating the parameter-compression penalty via Theorem 4.2 reduces the complexity of quadratic-form evaluations by a factor of $\Theta\big((r/d)^2\big)$.*

*Proof.* First, we construct and store the complete Fisher matrix $F(\theta) \in \mathbb{R}^{d \times d}$, which itself needs to store $d^2$ scalars, so it is $\Theta(d^2)$. Then, evaluate the quadratic form $\Delta\theta^\top F, \Delta\theta$, which can be completed in two steps: first, calculate $u = F, \Delta\theta$, involving $d$ inner product operations of length $d$, totaling $\Theta(d^2)$, and second, calculate the scalar $\Delta\theta^\top u$, which requires an additional $\Theta(d)$, and the total is still $\Theta(d^2)$. If it is further necessary to solve $F^{-1}$ or perform eigendecomposition, the time complexity of the corresponding classic algorithm is $\Theta(d^3)$.

Next, in the low-rank method, the original vector $\theta \in \mathbb{R}^d$ is first mapped to an $r$-dimensional subspace using truncated SVD (or randomized SVD). The main computation comes from the multiplication and addition of the $p \times r$ matrix, so the complexity is $\Theta(d, r^2)$. Then, in this subspace, the gradient Jacobian $J = \nabla_{\tilde\theta} \log \pi_\theta$ is calculated and an approximate Fisher matrix $\tilde F = J^\top J$ is formed. Each entry requires $d$ multiplications and additions, and the overall complexity is still $\Theta(d, r^2)$. Finally, evaluating the quadratic form $(\Delta\tilde\theta)^\top \tilde F, (\Delta\tilde\theta)$ only requires multiplication and addition of the $r \times r$ matrix and the length $r$ vector, with a complexity of $\Theta(r^2)$; if $\tilde F$ needs to be further decomposed, it will be $\Theta(r^3)$. Therefore, the overall complexity of the approximate method can be expressed as

$$C_{\text{approx}} = C_{\text{SVD}} + C_{\text{grad}} + C_{\text{approx}}^{(2)} = \Theta(p\, r^2 + r^3). \tag{28}$$

Dividing the complexity of the above two methods can get the speedup ratio. On the one hand, the original method $C_{\text{orig}}^{(2)} = \Theta(d^2)$, on the other hand, the approximate method $C_{\text{approx}}^{(2)} = \Theta(r^2)$. Thus, we get:

$$\frac{C_{\text{orig}}^{(\text{eig})}}{C_{\text{approx}}} = \frac{\Theta(p^3)}{\Theta(p\, r^2 + r^3)} = \Theta\big((p/r)^2/(1 + r/p)\big) \approx \Theta\big((p/r)^2\big). \tag{29}$$

Next, we prove that the approximation error caused by the above approximation is limited and controllable.

**Theorem D.3** *Assume that in each episode update, the parameter increment $\Delta\tilde{\theta}_k = \tilde{\theta}_k - \tilde{\theta}_{k-1}$ satisfies $\|\Delta\tilde{\theta}_k\| \leq B$, and has a uniform bound $M$ on the third-order derivative of any $\theta$. Let $\widehat{\Delta I}_k = (\Delta\tilde{\theta}_k)^\top F(\tilde{\theta}_k) \Delta\tilde{\theta}_k$ and $\Delta I_k = I(\tilde{\theta}_k; s_k) - I(\tilde{\theta}_{k-1}; s_{k-1})$ and estimate the Fisher matrix through $N_\tau$ independent sampling trajectories, and then take $K$ episodes for joint statistics. Then for any confidence level $\delta \in (0, 1)$, with probability at least $1 - \delta$ we have:*

$$\max_{1 \leq k \leq K} \left|\Delta I_k - \widehat{\Delta I}_k\right| \leq \frac{M}{6} B^3 + \sqrt{\frac{8\ln(4 \cdot 2^d/\delta)}{N_\tau K}} \tag{30}$$

*where $d = \dim(\tilde{\theta})$ is the parameter dimension, the first term $\frac{M}{6}B^3$ comes from the third-order remainder of the second-order Taylor expansion, and the second term is derived from the matrix Hoeffding inequality or Bernstein inequality following Matrix-Concentration theory [46].*

*Proof.* We decompose the potential error into two parts: (i) Taylor expansion remainder: approximate the true mutual information increment with a second-order Taylor expansion, and the remaining third-order term gives the $\frac{M}{6}B^3$ upper bound. (ii) Sampling/statistical error: use the matrix condensation inequality to give the spectral norm level upper bound on the deviation between the empirical Fisher and the true Fisher, and then get the second term from the quadratic property. Next, we discuss and analyze these two items in turn.

For the function $f(\theta) = \log\pi_\theta(z_k \mid s_k)$, at point $\tilde{\theta}_k$, do a second-order Taylor expansion along the direction $h = \Delta\tilde{\theta}_k$, and we have

$$f(\tilde{\theta}_{k-1}) = f(\tilde{\theta}_k) - \nabla f(\tilde{\theta}_k)^\top h + \tfrac{1}{2} h^\top \nabla^2 f(\tilde{\theta}_k) h - R_3, \tag{31}$$

Among the remainders $R_3$, for a certain $\xi$ is between $\tilde{\theta}_{k-1}$ and $\tilde{\theta}_k$. From $\|\nabla^3 f\| \leq M$ and $\|h\| \leq B$, we get

$$R_3 = \frac{1}{6} h^\top \left[\nabla^3 f(\xi)[h, h]\right] h \leq \frac{M}{6} \|h\|^3 = \frac{M}{6} B^3. \tag{32}$$

Thus, we have:

$$\left|\Delta I_k - (\Delta\tilde{\theta}_k)^\top F (\Delta\tilde{\theta}_k)\right| \leq \frac{M}{6} B^3. \tag{33}$$

Next, we turn to discuss the empirical Fisher's condensation error. Assume there are $K$ episodes in total, and each episode collects $N_\tau$ independent trajectories. Let $g_{j,k} = \nabla_{\tilde{\theta}_k} \log\pi_{\mu_0}\left(z_k^{(j)} \mid s_k\right) \in \mathbb{R}^d$, $j = 1, \ldots, N_\tau$, $k = 1, \ldots, K$., then the true Fisher information matrix can be expressed as $F = \mathbb{E}[g\,g^\top]$, $g \overset{\text{i.i.d.}}{\sim} \{g_{j,k}\}$, and the empirical Fisher is $\widehat{F} = \frac{1}{N_\tau K} \sum_{k=1}^{K} \sum_{j=1}^{N_\tau} g_{j,k}\, g_{j,k}^\top$, note $n = N_\tau K$.

Let the matrix corresponding to the $i$th sample be (flatten the double subscript to a single subscript) $X_i = g_{j,k}\, g_{j,k}^\top - F$ where $i = 1, \ldots, n$, obviously $\mathbb{E}[X_i] = 0$ and $\widehat{F} - F = \frac{1}{n} \sum_{i=1}^{n} X_i$. Applying matrix Hoeffding inequality, we obtain that: If $\{X_i\}$ is an independent symmetric matrix and $\mathbb{E}[X_i] = 0, \|X_i\| \leq R$, then for all $u \geq 0$ we have $\Pr\left(\|\sum_{i=1}^{n} X_i\| \geq u\right) \leq 2d \exp\left(-\frac{u^2}{8\,n\,R^2}\right)$. Apply this to $\sum_i X_i = n(\widehat{F} - F)$, we have:

$$\Pr\left(\|\, n(\widehat{F} - F)\| \geq u\right) \leq 2d \exp\left(-\frac{u^2}{8\,n\,R^2}\right) \tag{34}$$

Let $u = n\,t$, we get

$$\Pr\left(\|\widehat{F} - F\| \geq t\right) = \Pr\left(\|n(\widehat{F} - F)\| \geq n\,t\right) \leq 2\,d\,\exp\left(-\frac{n^2 t^2}{8\,n\,R^2}\right) = 2\,d\,\exp\left(-\frac{n\,t^2}{8\,R^2}\right). \tag{35}$$

If we assume that each gradient norm is restricted: $\|g_{j,k}\| \leq 1$, then $\|g_{j,k}g_{j,k}^\top\| \leq 1$ and $\|F\| \leq 1$, so $\|X_i\| \leq \|g_{j,k}g_{j,k}^\top\| + \|F\| \leq 2$, where $R = 2$. In order to be effective for both positive and negative sides, it is often multiplied by 2 before the above formula, and we get:

$$\Pr\left(\|\widehat{F} - F\| \geq t\right) = \Pr\left(\|n(\widehat{F} - F)\| \geq n\,t\right)$$
$$\leq 2\,d\exp\left(-\frac{n^2 t^2}{8\,n\,R^2}\right) = 2\,d\exp\left(-\frac{n\,t^2}{8\,R^2}\right) \tag{36}$$
$$\leq 4\,d\exp\left(-\frac{n\,t^2}{8\,R^2}\right) = \delta \implies t = R\sqrt{\frac{8}{n}\ln\frac{4\,d}{\delta}},$$

Replace $n = N_\tau K$, $R = 2$, and remember $d \to 2^d$ (according to the parameter dimension, we get $t = 2\sqrt{\frac{8}{N_\tau K} \ln\left(\frac{4 \cdot 2^d}{\delta}\right)} = \sqrt{\frac{8 \ln\left(4 \cdot 2^d/\delta\right)}{N_\tau K}}$.. Under the above event (with probability $\geq 1 - \delta$), for any vector $h$, we have $\left|h^\top (\widehat{F} - F) h\right| \leq \|h\|^2 \|\widehat{F} - F\| \leq \|h\|^2 t$, which is the typical property of controlling quadratic forms using the spectral norm.

Superimposing the Taylor remainder with the empirical Fisher statistical error, we get for each episode:

$$
\left|\Delta I_k - h_k^\top \widehat{F} h_k\right| \leq \underbrace{\frac{M}{6} B^3}_{\text{Taylor remainder}} + \underbrace{B^2 t}_{\substack{\text{sampling/statistics} \\ \text{error}}}
$$
(37)
$$
= \frac{M}{6} B^3 + B^2 \sqrt{\frac{8 \ln(2^d/\delta)}{N_\tau K}}.
$$

If the update amount is normalized (or assumed $B \leq 1$), it can be simplified to $\epsilon = \frac{M}{6} B^3 + \sqrt{\frac{8 \ln(2^d/\delta)}{N_\tau K}}$. Thus, we complete the proof of **Theorem D.3**.

Therefore, we can conclude that the computation method in **Theorem 4.2** significantly reduces computational complexity (**Theorem D.2**), while the approximation error (**Theorem D.3**) remains bounded, thereby supporting the practical computation of the proposed reward.

## D.2 Intuition behind the Proposed Reward

### D.2.1 Intuition behind the Fitting Information Gain

**How to interpret LLM reasoning from an information-theoretic perspective (fitting gain vs. uncertainty)** The reasoning process of an LLM can be viewed as inferring the correct answer $Y$ from input $X$. The more certain the model's prediction is, the better it "understands" the answer. Based on classical information theory [37, 3, 14], we can use conditional entropy $H(Y|X)$ to characterize the model's uncertainty about the output $Y$. If the model is sufficiently confident, its $H(Y|X)$ should be low; conversely, if it is uncertain or making a blind guess, $H(Y|X)$ should be high. Within this framework, the goal of inference is to gradually reduce $H(Y|X)$ until the correct answer is output. Importantly, while $H(Y|X)$ represents the relationship between the input and output, this process is controlled by the LLM (determined by the parameter $\theta$). Therefore, a more reasonable metric is: Given $\theta$, what is the model's uncertainty about $Y$? That is, we want $\theta$ to not only capture the input $X$ but also provide strong discrimination of the correct answer $Y$. Therefore, we use conditional mutual information $I(\theta; Y|X) = H(Y|X) - H(Y|X, \theta)$ to measure how much the known model parameters $\theta$ reduce the uncertainty about $Y$ given $X$. This ties rewards to meaningful reasoning progress, i.e., the reward for each episode depends on how much it helps the model understand the correct answer. Therefore, we define "fitting information gain as the reduction in uncertainty".

**How to characterize the gradual improvement of inference (introducing episode gain with $\theta_k$ and $\theta_{k-1}$)** In each episode $k$, the model updates its parameter state from $\theta_{k-1}$ to $\theta_k$ by observing the new reasoning step $z_k$. The key question is: Does this episode help the model better "understand" the answer? Therefore, we define fitting information gain of this episode as $\Delta I_k = I(\theta_k; Y|X) - I(\theta_{k-1}; Y|X)$. If this incremental gain is significant, it indicates that this episode has helped the model become more certain and effective. Here, $\theta_k$ and $\theta_{k-1}$ represent the posterior model parameters before and after learning the knowledge from episode $k$, estimated by the change in log-probability during the forward prediction process.

**Why the proposed metric corresponds to reducing uncertainty (approximating information gain with $J_r$)** Because directly calculating mutual information is too complex, we introduce an approximate metric $J_r(\cdot)$ to represent the model's predicted probability of the correct answer. This metric improves as the model's "confidence" increases. Therefore, the fitting gain is approximated as $\Delta I_k \approx J_r(\pi_\theta(\cdot \mid s_k, z_k)) - J_r(\pi_\theta(\cdot \mid s_k))$, with theoretical guarantees in Appendix D.1. This

difference measures whether episode $k$ improves the model's ability to predict the correct answer, reflecting the reduction process, i.e., the increasing reliability of reasoning.

### D.2.2 Intuition behind the Compression Penalty

**Why introducing compression penalty** In LLM reasoning, if each episode's information causes a significant change, while that episode may provide information gain, it may also capture unnecessary details within that episode, leading to overfitting or redundant computation. We want the model to only learn information that contributes to the answer within the episode. Therefore, inspired by the information bottleneck theory, we introduce a compression penalty based on the fitting term to further improve efficiency. It measures the "information overhead" incurred by each episode from an information-theoretic perspective.

**How is it measured (intuition behind the design)** If $\theta_k$ differs significantly from $\theta_{k-1}$, but the model's prediction performance (i.e., fitting gain) improves only slightly, this step may "absorb redundant information". Therefore, we use the mutual information increment $I(\theta_k; s_k) - I(\theta_{k-1}; s_{k-1})$ between them with the fitting term to measure whether the information in episode $k$ introduces unnecessary overhead.

**Why this is called compression** The term compression comes from the idea that a model should retain only the minimal amount of information sufficient to perform accurate reasoning. In our setting, the mutual information $I(\theta; s_k)$ quantifies how much the information of this episode is encoded into the model parameters $\theta$. A larger value implies that the model has to "store" more bits to fit that episode, akin to using more storage in a compressed file. By penalizing the increase $I(\theta_k; s_k) - I(\theta_{k-1}; s_{k-1})$, we explicitly discourage storing redundant information, promoting a more compact (i.e., compressed) internal representation. This aligns with principles from MDL and PAC-Bayes, where generalization is favored when the hypothesis (here, $\theta$) is simple and concise.

### D.3 More Comparison

Our framework advances prior and concurrent works in three key dimensions, i.e., efficiency, generality, and robustness, which we briefly illustrate below.

Firstly, previous methods [43, 16, 53, 9, 49] mainly optimize via outcome rewards, which drives models to over-extend reasoning chains and waste test-time compute. In contrast, our dense process rewards immediately quantify each episode's contribution to performance, enabling the model to learn when to stop reasoning and thus achieve equal or better accuracy with a minimal token budget.

Secondly, some concurrent works [57, 1, 33, 36, 20] propose task-specific process rewards, heuristic scorers, and length penalties to reduce the length of the reasoning chains, helping save test-time compute. However, these methods require costly manual labeling and do not transfer across tasks since they rely on task-specific settings, and there is no one-size-fits-all solution. We instead measure the parameter-update signal inspired by information theory, i.e., the intrinsic change in the model's weights after each optimization, as a task-agnostic proxy for learning progress. This internal metric requires no additional annotations or retraining and applies uniformly across diverse reasoning tasks.

Thirdly, existing reward-based updates to optimize test-time compute [36, 57, 1, 40, 64] may embed noise or task-specific artifacts into model weights, leading to overfitting and drastic performance drops under slight input shifts. In contrast, we introduce a parameter compression penalty that quantifies and negatively rewards the absorption of redundant information at each update. Only updates yielding true information gain are amplified; small or harmful directions are suppressed, ensuring stability under noisy, ambiguous, or adversarial inputs. Moreover, by recasting LLM fine-tuning as a episodic RL problem, combining a task distribution with dense per-step feedback, we enable the policy to maintain robust performance from simple arithmetic to complex proofs, without redesigning rewards or retuning hyperparameters for each new task.

Thus, by combining these three advantages into a unified RL objective, our L2T simultaneously maximizes reasoning effectiveness and computational efficiency across tasks of varying complexity, unlike prior approaches that trade off one for the other or rely on bespoke reward designs.

### D.4  Broader Impacts and Limitations

In this subsection, we briefly illustrate the broader impacts and limitations of this work.

**Broader Impacts.** This work explores how to simultaneously maximize inference effectiveness and efficiency across tasks of varying complexity to meet real-world demands. It reformulates LLM reasoning through a episodic reinforcement-learning and information-theoretic lens, and introduces a general dense process reward to track reasoning progress, enabling optimal performance under a minimal token budget. Extensive theoretical and empirical analyses validate its effectiveness and robustness. This work also opens up exciting new avenues for future research, e.g., provides a way for more explicit and automated dynamic budget allocation in the future, especially in scenarios sensitive to token costs (e.g., mobile deployment or real-time QA).

**Limitations.** This study evaluates general LLM reasoning tasks, such as mathematical proofs and code generation, which are commonly used to verify the reasoning capability of LLMs; some newly proposed benchmarks, such as web design, were not used. Our current experiments use mainly the open-source DeepSeek base models, including the scales of 1.5B, 3B, and 7B, while the scale of 72B and even above 100B is not used due to resource limitations and not being open source. We will investigate additional case studies and more base models to extend this work in the future.

## E  Benchmark Datasets

In this section, we briefly introduce all datasets used in our experiments. In summary, the benchmark datasets can be divided into two categories: (i) reasoning tasks for mathematical derivation, including AIME24-25, AMC, MATH500 [18], MinervaMATH [27], and Omni-MATH [13]; and (ii) reasoning tasks for code generation via HumanEval [8] The compositions of these benchmarks are as follows:

- AIME24-25 comprises 30 questions from the 2024 and 2025 American Invitational Mathematics Examination, with 15 fill-in-the-blank questions per exam. These questions are more difficult than AMC, spanning number theory, combinatorics, geometry, and algebra.

- AMC10/12 consists of 25 multiple-choice questions each for the AMC10 (up to 10th grade) and AMC12 (up to 12th grade). Each competition consists of 25 multiple-choice questions, totaling 975 questions across 39 tests. Questions progress from basic algebra and geometry to introductory probability and counting, covering various tasks for LLM reasoning evaluation.

- MATH500 is a 500-question subset randomly sampled from MATH, covering seven topics—prealgebra, algebra, number theory, geometry, intermediate algebra, precalculus, etc. Each question includes a step-by-step solution and a difficulty label from 1 to 5, enabling evaluation of an LLM's mathematical question-solving across diverse domains.

- MinervaMATH comprises 12,500 high-school-level contest questions. Each includes detailed solution steps and spans prealgebra through precalculus.

- Omni-MATH is an Olympiad-level benchmark of 4,428 competition questions across 33 subdomains (e.g., number theory, combinatorics, geometry, algebra), stratified into over 10 difficulty levels (divided into 4 tiers following [5]).

- HumanEval consists of 164 Python programming tasks designed to evaluate the correctness of code generated by models. Each task includes a standard function signature, and the model must generate the corresponding code implementation based on the description. The evaluation metric is primarily Pass@k, which measures the proportion of times the generated code passes the test cases at least once within k attempts.

## F  Implementation Details

For model training, we directly load the base models from Hugging Face, including DeepScaleR-1.5B-Preview, DeepSeek-R1-Distill-Qwen-1.5B, DeepSeekR1-Distill-Qwen-7B, and Qwen2-7B-Instruct. For different reasoning tasks, we introduce the experimental settings in the corresponding sections of **Section 5** and **Appendix G**. Unless otherwise specified, we follow the protocol of each benchmark and record the maj@4 results across different models. The training configuration is: the learning

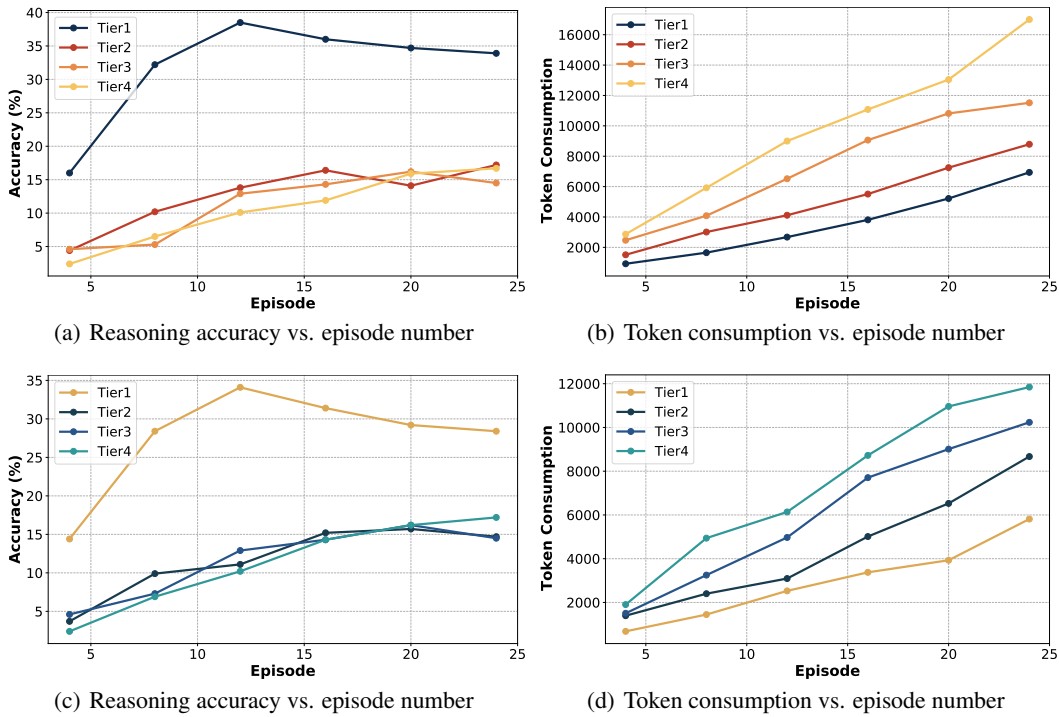

(a) Reasoning accuracy vs. episode number

(b) Token consumption vs. episode number

(c) Reasoning accuracy vs. episode number

(d) Token consumption vs. episode number

Figure 6: Results of DeepScaleR-1.5B-Preview (a,b) and DeepSeek-R1-Distill-Qwen-1.5B (c,d) across different tasks on Omni-MATH. We partition the generated reasoning chain into episodes, measuring accuracy $\text{Acc}(k)$ and average token consumption $\overline{T}(k)$ at different episode depths.

rate is set to $1.0 \times 10^{-6}$, with a cosine learning rate scheduler, and a warm-up ratio of 0.1. We use a batch size of 256, with a maximum prompt length of 4,096 tokens and a maximum completion length of 16,384 tokens. The model is trained for 1 epoch, up to 10 epochs. Additionally, we set the 'use_vllm' flag to True to enable vLLM acceleration, with a GPU memory utilization of 0.8. We also utilize mixed precision training with BF16 enabled. The parameters for compression penalty approximation are handled by a single-layer MLP, with a Fisher information matrix damping factor set to $10^{-5}$. The regularization hyperparameters $\alpha$ and $\beta$ are set to 0.8 and 0.6 according to grid search results, respectively. Also, $\alpha$ can be set to 1 for simplicity, with the performance drop less than 1%. More evaluation of implementation is provided in **Appendix G**, e.g., parameter sensitivity, prompt configuration, etc. The entire training is conducted on A100 GPU clusters, ensuring scalability and high computational efficiency.

## G    Additional Experiments and Full Results

In this section, we present the full results and additional experiments of this work, including extended settings, datasets, and base models, which are provided in the appendix due to space limitations.

### G.1    More Details and Results of the Motivating Experiments

In **Subsection 3.2**, we evaluate how efficiently existing methods use tokens. We benchmark two base models, DeepScaleR-1.5B-Preview and DeepSeek-R1-Distill-Qwen-1.5B, on Omni-MATH (4,428 questions across 33+ subfields, split into Tiers 1-4 by expert difficulty labels). Both models have been fine-tuned with outcome-based RL. To study performance at varying reasoning depths, we split each generated reasoning chain into up to $K = 20$ episodes using '<think>...</think>', then record the sequential-generation accuracy $\text{Acc}(k)$ and the average token usage $\overline{T}(k)$ at each episode $k$. For comparison, we also include a Maj4 baseline: under the same truncated context, we sample four answers and take a majority vote, measuring $\text{Maj@4}(k)$ and its token cost. Plotting "accuracy

Table 2: Pass@1 performance on various math reasoning benchmarks. We compare base models trained with different fine-tuning approaches. The best results are highlighted in **bold**.

| Base model + Method | AIME 2024 | AIME 2025 | AMC 2023 | MATH500 | MinervaMATH | **Avg.** |
|---|---|---|---|---|---|---|
| **DeepScaleR-1.5B-Preview** | 42.8 | 36.7 | 83.0 | 85.2 | 24.6 | 54.5 |
| +outcome-reward RL (GRPO) | 44.5 (+1.7) | 39.3 (+2.6) | 81.5 (-1.5) | 84.9 (-0.3) | 24.7 (+0.1) | 55.0 (+0.5) |
| +length penalty | 40.3 (-2.5) | 30.3 (-6.4) | 77.3 (-5.7) | 83.2 (-2.0) | 23.0 (-1.6) | 50.8 (-3.7) |
| +ReST-MCTS | 45.5 (+2.7) | 39.5 (+2.8) | 83.4 (+0.4) | 84.8 (-0.4) | 23.9 (-0.7) | 55.4 (+0.9) |
| +MRT | 47.2 (+4.4) | 39.7 (+3.0) | 83.1 (+0.1) | 85.1 (-0.1) | 24.2 (-0.4) | 55.9 (+1.4) |
| +Ours | **48.5 (+5.7)** | **40.2 (+3.5)** | **85.4 (+2.4)** | **88.1 (+2.9)** | **26.5 (+1.9)** | **57.8 (+3.3)** |
| **DeepSeek-R1-Distill-Qwen-1.5B** | 28.7 | 26.0 | 69.9 | 80.1 | 19.8 | 44.9 |
| +outcome-reward RL (GRPO) | 29.8 (+1.1) | 27.3 (+1.3) | 70.5 (+0.6) | 80.3 (+0.2) | 22.1 (+2.3) | 46.0 (+1.1) |
| +length penalty | 27.5 (-1.2) | 22.6 (-3.4) | 64.4 (-5.5) | 77.1 (-3.0) | 18.8 (-1.0) | 42.0 (-2.9) |
| +ReST-MCTS | 30.5 (+1.8) | 28.6 (+2.6) | 72.1 (+1.2) | 80.4 (+0.3) | 20.3 (+0.5) | 46.4 (+1.5) |
| +MRT | 30.3 (+1.6) | 29.3 (+3.3) | 72.9 (+3.0) | 80.4 (+0.3) | 22.5 (+2.7) | 47.1 (+2.2) |
| +Ours | **32.9 (+4.2)** | **30.1 (+4.1)** | **73.5 (+3.6)** | **84.7 (+4.6)** | **24.5 (+4.7)** | **49.2 (+4.3)** |
| **DeepSeek-R1-Distill-Qwen-7B** | 55.5 | 50.2 | 85.1 | 87.4 | 42.1 | 64.1 |
| +outcome-reward RL (GRPO) | 56.9 (+1.4) | 51.7 (+1.5) | 85.5 (+0.4) | 87.7 (+0.3) | 43.5 (+1.4) | 65.1 (+1.0) |
| +length penalty | 53.8 (-1.7) | 46.9 (-3.3) | 81.2 (-3.9) | 83.7 (-3.7) | 39.5 (-2.6) | 61.0 (-3.1) |
| +MRT-Reproduct | 57.0 (+1.5) | 52.4 (+2.2) | 86.0 (+0.9) | 88.4 (+1.0) | 44.3 (+2.2) | 65.6 (+1.5) |
| +Ours | **58.4 (+2.9)** | **53.6 (+3.4)** | **87.5 (+2.4)** | **89.2 (+1.8)** | **45.0 (+2.9)** | **66.8 (+2.7)** |

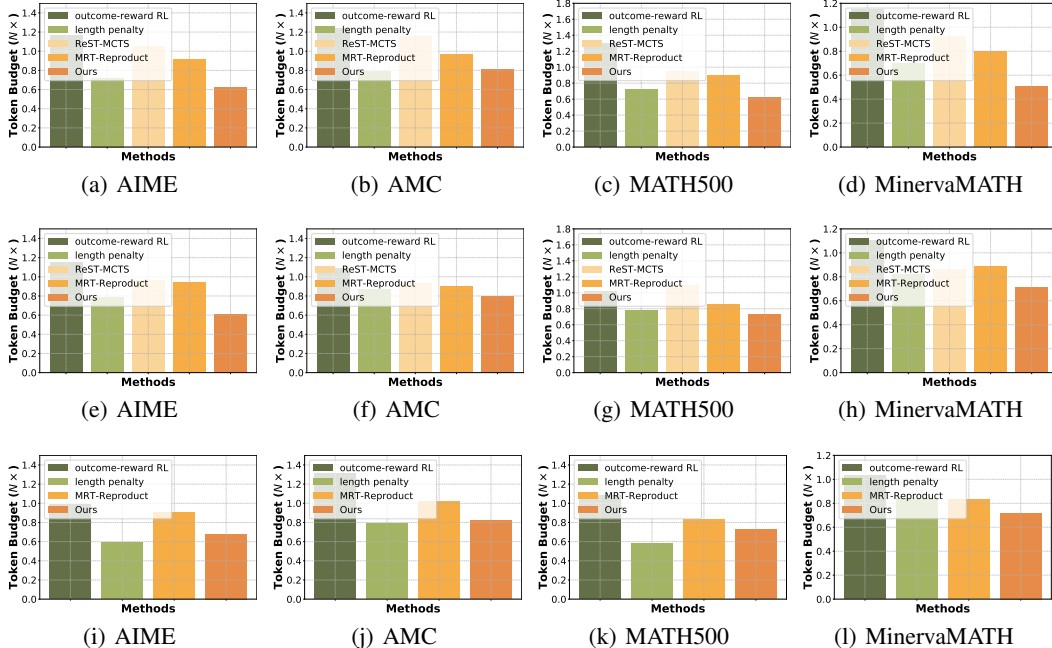

(a) AIME    (b) AMC    (c) MATH500    (d) MinervaMATH

(e) AIME    (f) AMC    (g) MATH500    (h) MinervaMATH

(i) AIME    (j) AMC    (k) MATH500    (l) MinervaMATH

Figure 7: Efficiency comparison on DeepScaleR-1.5B-Preview (a-d), DeepSeek-R1-Distill-Qwen-1.5B (e-h), and DeepSeek-R1-Distill-Qwen-7B (i-l). We compute the token budget required for each benchmark and treat the budget of the base model w/o fine-tuning as reference ($1\times$).

vs. episodes" and "token cost vs. episodes" reveals how model performance varies with question difficulty and compute budget, and highlights the relative merits of sequential decoding versus voting.

Due to space constraints, we previously showed only DeepScaleR-1.5B-Preview. **Figure 6** now presents both base models, demonstrating the same trends: (i) $\text{Acc}(k)$ peaks and then declines as $k$ increases, indicating extra episodes add no new information and may hurt performance via context redundancy; (ii) token usage rises rapidly with $k$, exceeding twice the minimal budget before peak accuracy, underscoring that existing methods may not efficiently use test-time compute; and (iii) the optimal $k$ depends on difficulty—Tier 4 questions benefit from longer chains, whereas Tier 1 questions achieve strong results with very few episodes. These findings motivate L2T's dense process reward, which adaptively adjusts reasoning depth.

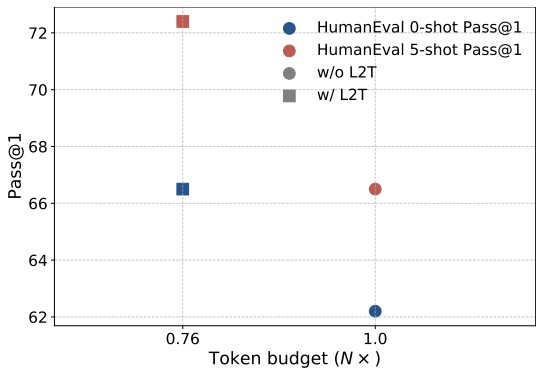

Figure 8: Effectiveness and efficiency analysis on code-related reasoning tasks.

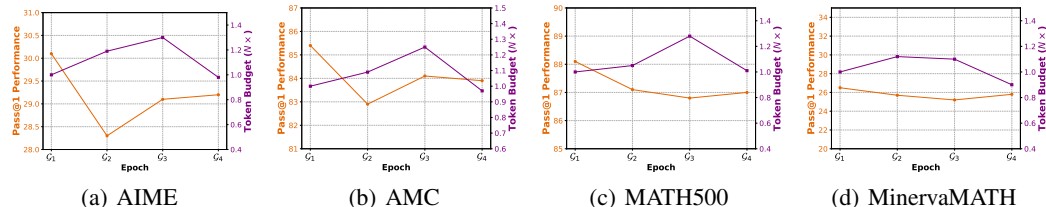

| (a) AIME | (b) AMC | (c) MATH500 | (d) MinervaMATH |

Figure 9: The effects of different components within L2T across different tasks.

## G.2 Full Results of Effectiveness and Efficiency Analysis

To evaluate the proposed L2T, we conduct experiments on various benchmarks, including mathematical and code-related tasks, and across base models of different scales. In the main text, we report the performance of L2T on mathematical reasoning tasks of varying complexity. **Table 1**, **Figure 2**, and **Figure 3** show that, compared to prior outcome-reward or process-reward methods, L2T delivers superior reasoning with less test-time compute. In this subsection, we assess its performance on more base models and more reasoning tasks, e.g., code generation tasks. Notably, mathematical reasoning and code generation serve as classic benchmarks for testing an LLM's complex reasoning ability [43, 16, 32]. First, we report performance across additional model scales. The results for inference accuracy and compute efficiency are shown in **Table 2** and **Figure 7**. We observe the same conclusion: L2T achieves state-of-the-art performance, attaining the highest inference accuracy with the smallest token budget. These findings demonstrate the broad effectiveness of our approach across models of varying scales. Secondly, we provide the performance of the proposed framework on the code generation task, and we evaluate the improvement of L2T on the LLM reasoning performance on the code generation task. Specifically, we run the GRPO/L2T fine-tuning pipeline on Qwen2-7B-Instruct and evaluate it using the standard HumanEval protocol. We set the temperature to 0.6 and top-p to 0.95 and generate 64 solutions per question, with a 1s timeout per attempt. We report the proportion of questions that pass all unit tests at least once. From the results in **Figure 8**, we can observe that compared with the outcome-reward-based RL method, L2T achieves better reasoning performance with less token budget. This proves the superiority of our proposed L2T and dense process rewards.

To further validate the versatility of our approach, we conduct two complementary studies. First, since L2T improves reasoning efficiency by adaptively allocating token budgets, we examine whether additional test-time search (e.g., MCTS) provides further gains. We apply MCTS to models fine-tuned with GRPO and L2T, and evaluate pass@1 accuracy on AIME and MinervaMATE. The results in **Table 3** show that GRPO benefits noticeably from MCTS, while L2T already achieves strong reasoning performance and gains only marginally, confirming that L2T effectively reduces reliance on large-scale search. Second, as our dense process reward is defined in a task-agnostic form, we also apply it to inference-only methods as a reranking signal. We evaluate DeepSeek-R1-Distill-Qwen-7B under a best-of-16 setting on MinervaMATH, comparing log-likelihood reranking, PRM-based reranking, and L2T reranking. The results in **Table 4** demonstrate that L2T achieves the highest accuracy, outperforming both likelihood-based and PRM baselines. Together, these findings indicate

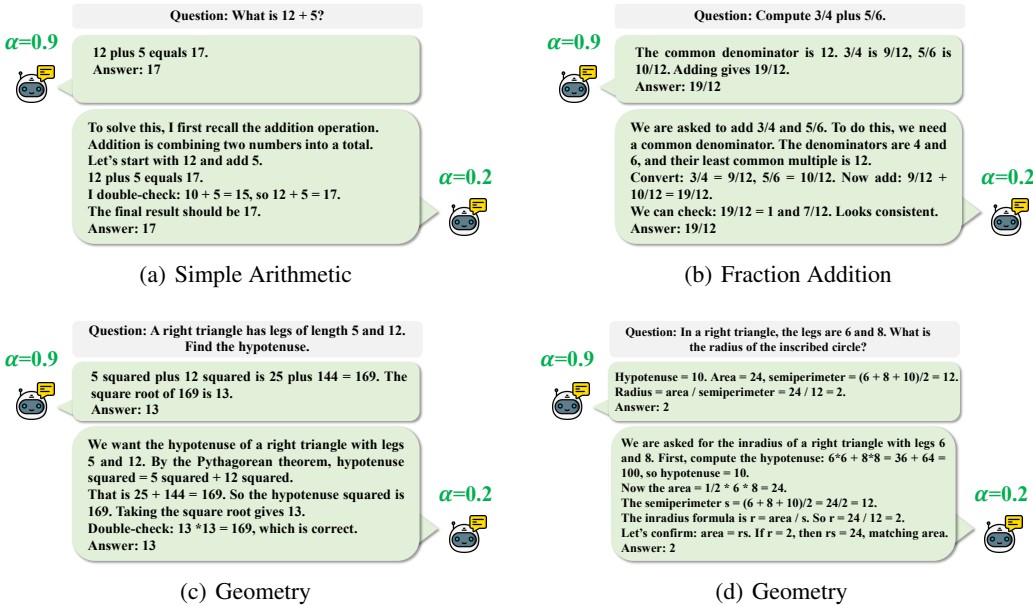

Figure 10: Examples of qualitative analysis about $\alpha$.

| Method | AIME 2024 | MinervaMATE |
|---|---|---|
| GRPO | 44.5 | 24.7 |
| GRPO+MCTS | 46.8 | 25.3 |
| L2T | 48.5 | 26.5 |
| L2T+MCTS | 48.9 | 26.7 |

Table 3: Effect of combining with MCTS.

| Method | MinervaMATH |
|---|---|
| Best-of-16 + log-likelihood | 42.6 |
| Best-of-16 + Qwen2.5-Math-PRM | 43.3 |
| Best-of-16 + Skywork-PRM | 43.0 |
| Best-of-16 + L2T rerank | 43.9 |

Table 4: Applying to inference-only reranking.

that L2T not only enables efficient reasoning during fine-tuning but also serves as an effective scoring mechanism in inference-time pipelines.

### G.3 Full Results of Ablation Studies

In **Subsection 5.3**, we conduct ablation studies to evaluate the optimal configuration and parameter settings. Considering questions of varying complexity, we perform evaluation on multiple benchmarks. We conduct an ablation study on the three core components of L2T, evaluating the contribution of each one by constructing alternative configurations: (i) Replacing information gain (config 1): replacing the proposed process reward with a task-aligned pre-trained reward model; (ii) Removing parameter compression penalty (config 2): completely removing the parameter compression penalty driven by the Fisher information matrix; (iii) Replacing low-rank approximation (config 3): using random sampling of 30% of network layers to approximate the Fisher information matrix, instead of the original low-rank approximation method. All ablation configurations follow the same hyperparameters and test protocols as the main experiment, each Tier is repeated five times to report average accuracy and average token consumption. The results in **Figure 9** confirm similar conclusions as in the main text: replacing information gain with task-specific reward leads to an average accuracy drop of about 1.9%, with a slight increase in token consumption; removing the parameter compression penalty results in about a 12% increase in consumption and a drop in accuracy; while random layer sampling reduces approximation overhead, the accuracy drops significantly, and the fluctuations increase substantially. These results validate the crucial role of each proposed component within L2T.

### G.4 Visualization

#### G.4.1 Qualitative Analysis of $\alpha$

In this subsection, we construct qualitative experiments to illustrate how to eliminate redundant inferences at low and high alpha values. The coefficient $\alpha$ denotes the weight of our proposed process reward (Eq.5). When $\alpha$ is small, the model relies more on the outcome reward, which provides high rewards only when the correct answer is found. Without guidance and given that correct answers are sparse in the output space, the model may consume a large number of tokens in exploration, reducing efficiency. In contrast, when $\alpha$ is large, the model is driven by the process reward, which assigns high rewards only if the current reasoning step has a high contribution to the accuracy of the answer, i.e., the correct answer is reached within a short token sequence. This encourages the model to generate informative tokens at each step, thereby improving efficiency. The qualitative results are shown in **Figure 10**, which demonstrate the above analyses. Take the Tier-1 Omni-MATH problem "What is $12 + 5$?" as an example, the qualitative analysis shows that with $\alpha = 0.9$, the model may answer within 2–3 steps, whereas with $\alpha = 0.2$, it may take more than 7 steps.

#### G.4.2 Prompt Configuration for Episode Segmentation

For episode segmentation, we automatically segment the chains by designing specific prompts.

Taking mathematical reasoning tasks as an example, to segment the reasoning chain into fixed episodes (e.g., "segment up to 30"), we add the following instruction to the prompt file:

```
<think>
In this section, show your detailed reasoning process.  Break down your
    reasoning into at most 30 logically coherent segments.

Each segment must be clearly marked with numbered tags in the format <episode_1>
    ... </episode_1>, <episode_2> ... </episode_2>, ..., up to <episode_30>.

Ensure each <episode_i> should contain only a single complete logical move, such
    as a definition, a formula derivation, a transformation, or a case split.
...
</think>
```

For adaptive segmentation (e.g., "break the reasoning chains into $K$ segments"), we add:

```
<think>
In this section, show your detailed reasoning process. Break down your reasoning
    into logically coheret segments.

Each segment should be enclosed in <episode_k> ... </episode_k> tags, such as:
<episode_1>...</episode_1>
<episode_2>...</episode_2>
...
<episode_K>...</episode_K>

Ensure each  <episode_i> should contain only a single complete logical move,
    such as a definition, a formula derivation, a transformation, or a case
    split. Avoid grouping multiple logical steps into one.

...
</think>
```

#### G.4.3 Visualization of Episodes and Dense Reward

In practice, episode segmentation is guided by semantic completeness and logical separability. It is automatically handled by the model during generation and sampling via prompt instructions, rather

than based on a fixed token count. As a result, episode lengths are variable. For example, each episode may correspond to a distinct reasoning action, such as "defining variables", "substituting into equations", or "structural inference". We provide a brief example below:

```
Q: Let x + y + z = 100, where x, y, z are positive integers such that gcd(x, y,
    z) = 1. How many such ordered triples (x, y, z) are there?

<think>
...

<episode_1>
We are given a positive integer equation x + y + z = 100 with the constraint
    that gcd(x, y, z) = 1. Our goal is to count the number of positive integer
    triples satisfying both conditions.
[Dense Reward: 0.03]
</episode_1>

<episode_2>
To simplify the counting, we observe that the total number of positive integer
    solutions to x + y + z = 100 is given by the standard stars-and-bars
    formula C(99, 2).
[Dense Reward: 0.08]
</episode_2>

<episode_3>
Now we need to subtract from this count the number of triples where gcd(x, y, z)
     > 1. To do this, we use the Mobius inversion principle to count only the
    primitive triples with gcd = 1.
[Dense Reward: 0.15]
</episode_3>

<episode_4>
Let d = gcd(x, y, z). Then we write x = da, y = db, z = dc so that a + b + c =
    100/d, and gcd(a, b, c) = 1.
[Dense Reward: 0.17]
</episode_4>

<episode_5>
We now compute the number of such primitive triples using the formula: sum over
    all d that divide 100 of mu(d) * C(100 / d - 1, 2), where mu(d) is the
    Mobius function. This ensures that we only count the triples with gcd = 1.
[Dense Reward: 0.23]
</episode_5>

...

<episode_8>
Therefore, the total number of ordered triples (x, y, z) such that x + y + z =
    100 and gcd(x, y, z) = 1 is 2960.
[Dense Reward: 0.12]
</episode_8>

<episode_9>
We briefly confirm the result by noting that the Mobius-based inclusion-
    exclusion method is standard and sound for counting primitive compositions.
[Dense Reward: 0.02]
</episode_9> </think>

...
```

To accommodate tasks with varying reasoning granularity: for problems with long reasoning chains (e.g., mathematical proofs or code generation), a single episode may span multiple tokens; in

contrast, for shorter or more fragmented tasks (e.g., factual questions), episodes may be automatically compressed into short logical units without forced segmentation. Below, we provide two examples.

A question with a short reasoning chain:

```
Q: Lily has 7 pencils. She buys 5 more and gives 3 to her friend. How many
    pencils does she have now?
...
<episode_1> Lily starts with 7 pencils. </episode_1>
<episode_2> She buys 5 more, so now she has 7 + 5 = 12 pencils. </episode_2>
<episode_3> She gives away 3 pencils, so 12 - 3 = 9 pencils remain. </episode_3>
```

A question with (relatively) long reasoning chains:

```
Q: Let a, b, and c be real numbers such that
a + b + c = 6,
ab + bc + ca = 9,
abc = 2.
Find a^3 + b^3 + c^3.
...
<episode_1> We are given a + b + c = 6, ab + bc + ca = 9, and abc = 2. </
    episode_1>
<episode_2> Recall the identity: a^3 + b^3 + c^3 = (a + b + c)^3 - 3(a + b + c)(
    ab + bc + ca) + 3abc. </episode_2>
<episode_3> Compute (a + b + c)^3 = 6^3 = 216. </episode_3>
<episode_4> Compute 3(a + b + c)(ab + bc + ca) = 3 * 6 * 9 = 162. </episode_4>
<episode_5> Compute 3abc = 3 * 2 = 6. </episode_5>
<episode_6> Substitute into the identity: 216 - 162 + 6 = 60. </episode_6>
```

