# OpenReview forum: "Learning to Think: Information-Theoretic Reinforcement Fine-Tuning for LLMs"
_NeurIPS.cc/2025/Conference — NeurIPS 2025 poster_

### Official Review · Reviewer_R3Rw · 2025-06-04

**Clarity:** 3
**Significance:** 3
**Originality:** 2
**Rating:** 4
**Confidence:** 4

**Summary:**

This paper presents Learning to Think (L2T), an information-theoretic reinforcement fine-tuning framework designed to enhance the reasoning effectiveness and efficiency of LLMs.
 L2T addresses the issue of unnecessary long reasoning chains in existing methods by introducing a universal dense process reward that uantifies episode-wise information gain, requiring no extra annotations or task-specific evaluators. The framework estimates this reward based on PAC-Bayes bounds and Fisher information matrix, theoretically reducing computational complexity while maintaining estimation accuracy.
Through experiments on various reasoning benchmarks and base models, L2T demonstrates advantages in both reasoning effectiveness and efficiency, adaptively allocating reasoning depth across different tasks to achieve optimal results with minimal token budgets.

**Questions:**

The paper utilizes two types of rewards: output reward for correctness and process reward aimed at efficiency.
Could the authors clarify how these two rewards are combined in the advantage estimation step? What specific advantage estimation technique is employed to jointly account for both the correctness and efficiency rewards when optimizing the policy?

**Ethical Concerns:**

["NO or VERY MINOR ethics concerns only"]

**Final Justification:**

4: Borderline accept: Technically solid paper where reasons to accept outweigh reasons to reject, e.g., limited evaluation. Please use sparingly.

The author has addressed my concerns.

**Limitations:**

yes

**Paper Formatting Concerns:**

No concerns

**Quality:**

3

**Strengths And Weaknesses:**

# Strengths
1. This paper propose a novel framework for efficient chain-of-thought reasoning from the perspectives of information theory based reinforcement learning.

2. This paper provides theoretical basis for the proposed dense process rewards, using PAC-Bayes bounds and Fisher information matrix for estimation.

3. This paper conducts extensive experiments across multiple reasoning benchmarks, and the experimental results show that L2T improves the efficiency and performance.


# Weaknesses
1. The paper focuses on efficient chain-of-thought reasoning.
To enhance the comprehensiveness of the experimental validation, the authors are encouraged to compare their approach with more related works that leverage reinforcement learning to improve reasoning efficiency.

2. The writing in Sections 3 and 4 is somewhat disorganized, with extensive use of in-line formulas making it difficult to read. It is recommended to clarify these sections. Moreover, presenting the overall methodological framework using a schematic diagram would help improve clarity and reader understanding.

---

> ### Author Rebuttal · Authors · 2025-07-29
>
> # Responses for Reviewer R3Rw
>
> We sincerely appreciate the reviewer R3Rw's constructive suggestions and the time and effort dedicated to the review process. We are also grateful for the recognition of our work and sincerely hope the following responses can help address the concerns.
>
>
>
> ## **Response to Weakness 1**
>
>
> - We have compared L2T with *several representative RL-based methods that focused on efficient reasoning and were released before the NeurIPS 2025 deadline*, such as ReST-MCTS [62] (Nov 2024), length penalty [2] (Feb 2025), and MRT [36] (Mar 2025). All comparisons were conducted under the same experimental setup and evaluation protocol. The results in Table 1–2 and Figures 2, 3, and 7 demonstrate the superiority of L2T, i.e., compared to the SOTA baselines, it achieves over 2% performance improvement while using less than 70% of the tokens.
> - According to the reviewer's valuable suggestions, we further added more comparisons with recent works that focus on "efficiency" (publicly released or reported results under the same evaluation protocol). The table below shows results against methods published after early 2025, showing that L2T achieves best performance with a smaller token budget. All the results are added in the revised version. In addition, we have also compared L2T combined with PRM on inference-only baselines. The results further confirm the advantage of L2T (please see Response to Q2 for Reviewer NSuU). We are also comparing with some contemporaneous work (published after Jun 2025). Due to the time constraints of the rebuttal, the results on all the base models will be supplemented in the final version.
>
> | Model                         | Method         | Date | AMC  | Avg. Token Cost |
> | ----------------------------- | -------------- | ---------------------- | ---- | --------------- |
> | DeepSeek-R1-Distill-Qwen-1.5B | length penalty | Feb 2025               | 64.4 | 1x              |
> | DeepSeek-R1-Distill-Qwen-1.5B | TERL           | Apr 2025               | 69.8 | 1.4x            |
> | DeepSeek-R1-Distill-Qwen-1.5B | VRDecouple     | Apr 2025               | 72.4 | 1.7x            |
> | DeepSeek-R1-Distill-Qwen-1.5B | RL-STAR        | Apr 2025               | 71.2 | 2.1x            |
> | DeepSeek-R1-Distill-Qwen-1.5B | MRT            | Mar 2025               | 72.9 | 1.6x            |
> | DeepSeek-R1-Distill-Qwen-1.5B | L2T            | Ours                   | 73.5 | 1.1x            |
>
>
>
> ## **Response to Weakness 2**
>
>
> - According to the reviewer's valuable suggestions, we have carefully revised and restructured the in-line formulas in Sections 3 and 4. Specifically, the previously in-line formulas (L267, L269, L271) have been reformatted as standalone numbered equations, each accompanied by concise contextual comments for clarity.
> - Following the constructive comments, we have also added a framework diagram to illustrate the actual workflow of L2T (added in Appendix B; the section title has been updated from "Pseudo-Code" to "Pseudo-Code and Overall Procedure"). Since visual figures cannot be included during the rebuttal phase, we provide the following outline for reference:
>   - **Sampling & Generation**: We randomly sample a question $x$ from the training set and use the old policy $\pi_{\theta_{\text{old}}}$ to generate $G$ reasoning rollouts. Each rollout is automatically segmented into episodes using the `<think>` delimiter defined in the system prompt, allowing different logical fragments to be evaluated independently.
>   - **Reward Computation**: For each rollout, the reward $R_{i,k}$ consists of two parts. The first is a sparse outcome reward, i.e., $r^{\text{out}} = \mathbb{1}[z_{1:K} \text{ leads to correct } y^*]$. The second is the proposed dense process reward, i.e., $r_k^{\text{prg}} = \Delta I_k - \beta C_k$, where the information gain term is $\Delta I_k = J_r(\pi_\theta(\cdot \mid s_k, z_k)) - J_r(\pi_\theta(\cdot \mid s_k))$, and the compression penalty is $C_k = I(\theta_k; s_k) - I(\theta_{k-1}; s_{k-1})$  (Please see Section 4.2 and Appendix C.1 for details). Then, the reward for episode $k$ can be expressed as $R_{i,k} = r_i^{\text{out}} / K_i + \alpha r^{\text{prg}}_{i,k}$.
>   - **Advantage Estimation**: Following GRPO, we use token-level log-probability surprise as the weighting signal. Each $R_{i,k}$ is distributed over tokens using weights $w_{i,t} \propto -\log p_{\theta_{\text{old}}}(z_{i,t} \mid s_{i,t})$, resulting in per-token reward $r_{i,t}$. The truncated mean of these rewards (95%) gives $\tilde{r}\_i$. We then compute the group-level advantage as a z-score, i.e., $\hat{A}\_i = (\tilde{r}\_i - \bar{\tilde{r}}) / \sigma\_{\tilde{r}}$,
>     and rescale it to each token using its relative contribution, i.e., $A_{i,t} = \hat{A}\_i \cdot (r\_{i,t} / \tilde{r}\_i)$.
>   - **Optimization**: The final advantage $A_{i,t}$ is used in GRPO’s clipped policy gradient, with an additional KL penalty term to stabilize training.
>
> In summary, L2T is implemented on top of GRPO, with the key differences being the explicit construction of episodes and the integration of the proposed universal information-theoretic dense process reward.
>
>
>
> ## **Response to Question 1**
>
> In the above **Response to Weakness 2**, we provide a formal illustration about the reward composition and advantage estimation of L2T. Building upon the above response, we would like to provide an outline to further answer each point raised in the "Questions":
>
> 1. **The combination of the rewards for the following advantage estimation:**
>    - **Implementation:** The two rewards are first combined at the episode level, i.e., the outcome reward $r^{\text{out}}$ is uniformly distributed across $K_i$ episodes, while the process reward $r^{\text{prg}}\_k$, which measures information gain and includes a compression penalty, is scaled by a weight $\alpha$. The total episode reward is $R_{i,k} = r_i^{\text{out}} / K_i + \alpha r^{\text{prg}}\_{i,k}$. This reward is then distributed to each token within the episode, weighted by the token’s surprise.
>    - **Benefit to Effectiveness and Efficiency:** (i) This design considers both dense progress and global correctness, encourages the model to learn the most efficient and effective steps; (ii) The process reward encourages the model to focus on episodes that provide the most useful information while penalizing redundant ones; (iii) The reward relies on internal model signals (information gain), making it broadly applicable without external annotations.
>
> 2. **The specific advantage estimation for policy optimization:**
>    - **Implementation:** L2T adopts a GRPO-style advantage estimator that combines normalization and surprise-based token weighting. Episode rewards $R_{i,k}$ are first allocated to tokens based on their negative log-probabilities $w_{i,t} \propto -\log p_{\theta_{\text{old}}}(z_{i,t} \mid s_{i,t})$, forming token-level rewards $r_{i,t}$. A 95% truncated mean across rollouts is then z-normalized to obtain a group-level advantage $\hat{A}\_i = (\tilde{r}\_i - \bar{\tilde{r}}) / \sigma\_{\tilde{r}}$, which is redistributed to each token proportionally to its contribution, i.e., $A_{i,t}$.
>    - **Benefit to Effectiveness and Efficiency:**  This method ensures that each token receives a reward signal reflecting both its local importance and overall trajectory quality, guiding the model to prioritize tokens with the highest expected performance gain.
>
> 3. **Empirical Evidence:** Experiments in Sections 5.2 and 5.3 show that the token efficiency of L2T can be nearly doubled, while still outperforming other methods in accuracy (Table 1-2, Figures 2, 3, and 7); the proposed dense reward and advantage estimation together lead to more efficient and accurate reasoning under limited compute budgets (Figures 3, 4, and 9).
>
> In summary, based on the above process, the model can adaptively adjust reasoning depth within a limited budget, generating tokens with the highest contribution, thus achieving higher accuracy with fewer tokens.
>
> We have supplemented the above illustrations in the appendix and further summarized them in the main text based on the valuable suggestions.

---

> > ### Comment · Reviewer_R3Rw · 2025-08-09
> >
> > We thank the authors for their detailed response.
> > The supplementary experiments and clarifications provided in the revised manuscript have satisfactorily resolved the previously raised issues.
> > Therefore, I will raise the rating to 4.

---

> > > ### Author Response · Authors · 2025-08-09
> > >
> > > Dear Reviewer R3Rw,
> > >
> > > We sincerely appreciate your positive feedback and for recognizing the contribution of our work. It is a great honor that we were able to address all your concerns.
> > >
> > > Best wishes,
> > >
> > > The authors.

---

> ### Author Response · Authors · 2025-08-08
> **Friendly Reminder: Deadline Approaching**
>
> Dear Reviewer R3Rw,
>
> Thank you so much for your time and valuable feedback for reviewing. We truly appreciate your expertise and understand you may be very busy.  In the previous stage, we have improved our work based on your constructive suggestions, including: incorporating more baselines beyond the current comparisons, adjusting three inline formulas to be displayed on separate lines, and supplementing a framework diagram.
>
> As the discussion deadline is in less than 36 hours and no discussion yet, we would like to kindly check whether you had a chance to review our responses and whether there are any remaining concerns we could address. We’re happy to provide clarification or discuss further. Thank you again for your essential contribution.
>
> Best regards,
>
> The authors.

---

> > ### Comment · Area_Chair_5dBx · 2025-08-09
> >
> > Dear Reviewer @R3Rw,
> >
> > Given less than 24 hours remain in the discussion period, the authors and Area Chairs would be grateful if you could take a careful look at the authors' rebuttal. Hopefully the authors' extensive rebuttal addressed your initial concerns. Please let us know. Thank you !!!

---

### Official Review · Reviewer_NSuU · 2025-06-24

**Clarity:** 3
**Significance:** 3
**Originality:** 3
**Rating:** 4
**Confidence:** 4

**Summary:**

The authors propose a reinforcement learning framework for training LLMs to achieve optimal reasoning using a dense process reward designed based on an information theoretic approach. Treating each query-response interaction as a series of multiple episodes, the authors define an episode-wise information gain to estimate the contribution of each reasoning step of LLMs in achieving the desired output. This dense reward model based on PAC-Bayes bounds and the Fisher information is then used to optimize LLMs to encourage efficient reasoning. Experiments demonstrate that fine-tuning LLMs with the proposed dense reward model achieves superior results than standard outcome rewards.

**Questions:**

- Q. How is the length of each episode determined in practice? Does it make sense for tasks in which reasoning steps vary in length?
- Q. Can the proposed dense reward model used for inference-only methods (best-of-N, beam search, etc.)? How does that perform compared to using other process reward models?

**Ethical Concerns:**

["NO or VERY MINOR ethics concerns only"]

**Final Justification:**

I appreciate the authors' responses with additional experimental results and discussions. It is useful to see that the proposed method combined with test-time inference techniques results in further gains compared to baselines such as GRPO. The additional qualitative examples also better illustrate how models trained with the method perform in terms the length and type of output reasoning traces. While the method is theoretically justified and supported with empirical results, more analysis on why RL with an outcome-based reward model (GRPO) often significantly underperform compared to the proposed method would enhance empirical validation. I'm also a bit concerned with the results in Table 1 where the models become considerably worse after training with GRPO (e.g., DeepScaleR-1.5B-Preview on AMC 2023) and wonder whether a sufficient hyperparameter has been conducted for the baselines -- and if so, what are some plausible explanations of these negative results. Overall, I'm quite happy with research in this direction of designing dense reward signals without explicit training of process reward models and feel the original rating is still fair (but I've increased my confidence score).

**Limitations:**

Yes.

**Paper Formatting Concerns:**

None.

**Quality:**

3

**Strengths And Weaknesses:**

Strengths
- The proposed dense reward is theoretically well-motivated.
- It has the practical advantages of requiring no human annotations compared to other process reward models.
- Uses both fine-tuning approaches (outcome reward, length penalty) and inference methods (MCTS) as baselines.

Weaknesses
- More ablations that would be useful: 1) experiments with models of sizes other than 1.5B -- this would help better understand how the method scales, 2) combining fine-tuning with test-time approaches, e.g., MCTS -- this would help understand whether reasoning efficient fine-tuning avoids the need for test-time compute, or further gains can be achieved.
- Some qualitative analysis on the dense reward would have been nice to get a better sense of whether the reward model assigns sensible scores to episodes based on the episodes' contributions.

---

> ### Author Rebuttal · Authors · 2025-07-29
>
> # Responses for Reviewer NSuU
>
> We sincerely appreciate the reviewer NSuU's constructive suggestions and the time and effort dedicated to the review process. We are also grateful for the recognition of our work and sincerely hope the following responses can help address the concerns.
>
>
>
> ## **Response to Weakness 1**
>
>
> - We would like to kindly clarify that Appendix F.2 presents experiments with models other than 1.5B, e.g., DeepSeek-R1-Distill-Qwen-7B and Qwen2-7B-Instruct. The results in Table 2, Figure 7, and Figure 8 demonstrate the superiority of L2T, achieving over 2% performance improvement while using less than 60% of the tokens compared to SOTA methods.
> - We sincerely thank the reviewer for the valuable suggestions and further conducted toy experiments for evaluation. The following results and more details are also added in Appendix F.5 according to the constructive comments. Briefly, we integrated MCTS into models trained with L2T and GRPO, and evaluated their performance. The results below show that L2T already exhibits strong reasoning capabilities; MCTS brings only marginal improvement for L2T, while GRPO benefits more from MCTS. This further indicates that L2T effectively reduces reliance on large-scale search by guiding the model to generate the most informative tokens.
>
> | Method    | AIME | MinervaMATE |
> | --------- | --------- | ----------- |
> | GRPO      | 44.5      | 24.7        |
> | GRPO+MCTS | 46.8      | 25.3        |
> | L2T       | 48.5      | 26.5        |
> | L2T+MCTS  | 48.9      | 26.7        |
>
>
>
> ## **Response to Weakness 2**
>
> We sincerely appreciate the constructive suggestions and have further added more qualitative analyses in Appendix F.3 and Appendix F.4 as suggested.
>
> Specifically, we randomly sampled problems from different datasets and visualized the rewards along their reasoning trajectories. An example is shown below. The results indicate that L2T assigns relatively low rewards to non-essential steps (e.g., background information) that may not influence the model’s decision, while assigning higher rewards to critical steps such as final equations, structural counting, and Möbius techniques.
>
> Combined with the quantitative results in Tables 1–2, this further demonstrates the effectiveness of L2T.
>
> ```
> Q: Let x + y + z = 100, where x, y, z are positive integers such that gcd(x, y, z) = 1. How many such ordered triples (x, y, z) are there?
>
> <think>
> In this section, we analyze the problem step-by-step using 9 logically structured episodes.
>
> <episode_1>
> We are given a positive integer equation x + y + z = 100 with the constraint that gcd(x, y, z) = 1. Our goal is to count the number of positive integer triples satisfying both conditions.
> [Dense Reward: 0.03]
> </episode_1>
>
> <episode_2>
> To simplify the counting, we observe that the total number of positive integer solutions to x + y + z = 100 is given by the standard stars-and-bars formula C(99, 2).
> [Dense Reward: 0.08]
> </episode_2>
>
> <episode_3>
> Now we need to subtract from this count the number of triples where gcd(x, y, z) > 1 . To do this, we use the Möbius inversion principle to count only the primitive triples with gcd = 1.
> [Dense Reward: 0.15]
> </episode_3>
>
> <episode_4>
> Let d = gcd(x, y, z). Then we write x = da, y = db, z = dc so that a + b + c = 100/d, and gcd(a, b, c) = 1.
> [Dense Reward: 0.17]
> </episode_4>
>
> <episode_5>
> We now compute the number of such primitive triples using the formula: sum over all d that divide 100 of mu(d) * C(100 / d - 1, 2), where mu(d) is the Möbius function. This ensures that we only count the triples with gcd = 1.
> [Dense Reward: 0.23]
> </episode_5>
>
> ...
>
> <episode_8>
> Therefore, the total number of ordered triples (x, y, z) such that x + y + z = 100 and gcd(x, y, z) = 1 is 2960.
> [Dense Reward: 0.12]
> </episode_8>
>
> <episode_9>
> We briefly confirm the result by noting that the Möbius-based inclusion-exclusion method is standard and sound for counting primitive compositions.
> [Dense Reward: 0.02]
> </episode_9> </think>
>
> ...
> ```
>
>
>
> ## **Response to Question 1**
>
>
> * In practice, episode segmentation is guided by semantic completeness and logical separability. It is automatically handled by the model during generation and sampling via prompt instructions, rather than based on a fixed token count. As a result, episode lengths are variable. For example, each episode may correspond to a distinct reasoning action, such as "defining variables", "substituting into equations", or "structural inference". We provide a brief example in Response to Weakness 2 (the above response).
>
> * To accommodate tasks with varying reasoning granularity: for problems with long reasoning chains (e.g., mathematical proofs or code generation), a single episode may span multiple tokens; in contrast, for shorter or more fragmented tasks (e.g., factual questions), episodes may be automatically compressed into short logical units without forced segmentation. Below, we provide two examples. More examples and visualization results have been added in Appendix F.4, according to the constructive comments.
>
> A question with a short reasoning chain:
>
> ```
> Q: Lily has 7 pencils. She buys 5 more and gives 3 to her friend. How many pencils does she have now?
> ...
> <episode_1> Lily starts with 7 pencils. </episode_1>
> <episode_2> She buys 5 more, so now she has 7 + 5 = 12 pencils. </episode_2>
> <episode_3> She gives away 3 pencils, so 12 - 3 = 9 pencils remain. </episode_3>
> ```
>
> A question with (relatively) long reasoning chains:
>
> ```
> Q: Let a, b, and c be real numbers such that
> a + b + c = 6,
> ab + bc + ca = 9,
> abc = 2.
> Find a^3 + b^3 + c^3.
> ...
> <episode_1> We are given a + b + c = 6, ab + bc + ca = 9, and abc = 2. </episode_1>
> <episode_2> Recall the identity: a^3 + b^3 + c^3 = (a + b + c)^3 - 3(a + b + c)(ab + bc + ca) + 3abc. </episode_2>
> <episode_3> Compute (a + b + c)^3 = 6^3 = 216. </episode_3>
> <episode_4> Compute 3(a + b + c)(ab + bc + ca) = 3 * 6 * 9 = 162. </episode_4>
> <episode_5> Compute 3abc = 3 * 2 = 6. </episode_5>
> <episode_6> Substitute into the identity: 216 - 162 + 6 = 60. </episode_6>
> ```
>
>
>
> ## **Response to Question 2**
>
> Yes, our proposed dense reward can be applied in inference-only settings to score and rerank candidate reasoning paths.
>
> Based on the model DeepSeek-R1-Distill-Qwen-7B (also used in our paper, as shown in Appendix E and F.2), we conducted a toy experiment to rerank multiple reasoning paths. The results in the table below demonstrate the effectiveness of the proposed L2T, i.e., L2T reranking achieves the highest accuracy compared to other methods and PRMs. We sincerely thank the reviewers for the valuable suggestions, and all results are now added to the appendix of the paper.
>
> Furthermore, we have compared L2T with other process-reward-based methods such as MRT and ReST-MCTS in Section 5. The results (Table 1-2, Figures 2, 3, and 7) show that L2T consistently outperforms them, achieving better performance with fewer tokens across multiple datasets.
>
> | Model                       | Method                          | MinervaMATH |
> | --------------------------- | ------------------------------- | ----------- |
> | DeepSeek-R1-Distill-Qwen-7B | Best-of‑N@16 + log-likelihood   | 42.6        |
> | DeepSeek-R1-Distill-Qwen-7B | Best-of‑N@16 + Qwen2.5-Math-PRM | 43.3        |
> | DeepSeek-R1-Distill-Qwen-7B | Best-of‑N@16 + Skywork-PRM      | 43.0        |
> | DeepSeek-R1-Distill-Qwen-7B | Best-of‑N@16 + L2T rerank       | 43.9        |

---

> > ### Comment · Reviewer_NSuU · 2025-08-05
> >
> > Thank you for the additional results and clarifications.
> >
> > Reviewing the results reported in Table 1, I'd like to ask whether the authors have some explanations as to why RL with an outcome-based reward model (GRPO) often significantly underperform compared to the proposed method. For example, what are some reasons to believe that the information-theoretic dense reward leads to not only more efficient reasoning but also better accuracy than, e.g., GRPO which is specifically trained to optimize for accuracy. Also, I'm curious why the models become considerably worse after training with GRPO (e.g., DeepScaleR-1.5B-Preview on AMC 2023), when the algorithm optimizes for accuracy.

---

> > > ### Author Response · Authors · 2025-08-05
> > >
> > > We sincerely appreciate the reviewer’s response and the opportunity to further clarify our work. Below, we respond to the two raised points one by one, and sincerely hope that this will help alleviate the concerns.
> > >
> > > ### Regarding the advantage of L2T
> > >
> > > > Q: I'd like to ask whether the authors have some explanations as to why RL with an outcome-based reward model (GRPO) often significantly underperforms compared to the proposed method. For example, what are some reasons to believe that the information-theoretic dense reward leads to not only more efficient reasoning but also better accuracy than, e.g., GRPO which is specifically trained to optimize for accuracy.
> > >
> > > Yes, we have discussed the advantages of L2T and would like to clarify from the following three points：
> > >
> > > - **How the reward mechanism of L2T guides model learning:** L2T introduces an information-theoretic dense reward, enabling the model to evaluate whether each reasoning step contributes to the correctness of the answer. Only steps that significantly improve answer accuracy receive high rewards, while redundant steps yield the lowest rewards. This dense feedback mechanism enables the model to generate useful tokens at every step, rather than blindly extending reasoning chains. By maximizing the gain of each step, L2T encourages the model to achieve optimal results with minimal reasoning depth.
> > > - **The advantage under token budget constraints:** When token budgets are limited, L2T’s reward signal ensures that the model does not waste tokens on low-contribution steps. To maximize rewards within the given budget, the model learns to prioritize reasoning steps that have a substantial impact on the answer correctness, while discarding or simplifying steps with low information gain. This information-driven generation ensures that the model completes high-gain reasoning steps first, avoiding reasoning interruptions due to token exhaustion, and thereby maintaining reasoning effectiveness under strict token constraints.
> > > - **Comparison with GRPO and corresponding improvements:** GRPO uses the final result as the reward signal, which can improve overall model accuracy. However, due to the lack of feedback on intermediate steps and the sparsity of correct answers in the output space, the model often resorts to extending reasoning chains to ensure coverage of correct paths. As demonstrated in Section 3.2, under limited token budgets, in simple tasks, this may introduce excessive redundant steps; while in complex tasks, this may cause premature truncation of critical reasoning steps. To address this, L2T incorporates a dense process reward that allows the model to assess the value of each reasoning step during generation, dynamically allocating its limited budget to steps with the highest information gain. This mitigates redundant reasoning and resource waste, enabling better performance within constrained budgets.
> > >
> > > Therefore, as shown in Table 1, L2T can achieve better results than GRPO.
> > >
> > > ### Regarding the experiments
> > >
> > > > Q: Also, I'm curious why the models become considerably worse after training with GRPO (e.g., DeepScaleR-1.5B-Preview on AMC 2023), when the algorithm optimizes for accuracy.
> > >
> > > We would like to kindly clarify that we did not perform training and evaluation on each individual benchmark but **followed the experimental setting of [36], where base models are fine-tuned on two fixed training datasets and evaluated across multiple benchmarks (Section 5.1, L341-345)**, e.g., "DeepScaleR-1.5B-Preview...fine-tuning on the 919 AIME questions..., and DeepSeek-R1-Distill-Qwen-1.5B is fine-tuned on a random sample of 4,000 question-answer pairs from NuminaMath." Therefore, due to domain differences between datasets, it is possible that after fine-tuning, the performance of model trained on AIME may slightly drop on AMC.
> > >
> > > **All our results on GRPO are consistent with those reported in [36].** Importantly, after applying L2T, the model achieves consistent and significant improvements across all datasets, indicating that L2T effectively guides the model to prioritize reasoning steps with the greatest impact on final accuracy.

---

### Official Review · Reviewer_Sp46 · 2025-06-30

**Clarity:** 3
**Significance:** 4
**Originality:** 3
**Rating:** 5
**Confidence:** 4

**Summary:**

The paper introduces a general-purpose dense reward framework for Reinforcement Learning in LLMs to improve their reasoning abilities. The purpose of their framework is to avoid wasteful tokens that do not add value to the final prediction, improving not only efficiency but also performance by removing clutter from the context of the model. They show superior performance and efficiency compared to other contemporary approaches.

**Questions:**

Addressing the weaknesses mentioned above should be feasible in the rebuttal, so I expect to see the new descriptions (or an explanation as to what is needed to address my confusion) that will appear in the paper.

**Ethical Concerns:**

["NO or VERY MINOR ethics concerns only"]

**Final Justification:**

There were not major points of contention for me in the paper, just some clarifications required to improve readability. The authors have adequately addressed any concerns, and I did not find any major weaknesses unaddressed from other reviewers, so the score can remain unchanged.

**Limitations:**

The authors have not adequately discussed the limitation of having to select a token budget during training. Ideally, the token budget would emerge organically from training or from the rewards themselves, and would be different for problems of different difficulty. While not a weakness per se, it would be good to have a discussion of the issues and potential avenues for future research.

**Quality:**

3

**Strengths And Weaknesses:**

Strengths:
- The paper is simple to follow in retrospect and intuitively makes sense as to why anyone would want to apply similar methods, which is a major strength of the paper.
- It studies an important problem and proposes an efficient solution that has a good chance of having an impact in the community.

Weaknesses:
- Confusion with current notation and terminology. Examples: L120 $B_{token}$ and $T_{max}$, some terms defined in page 3 and appear again in page 6, making reading the paper a bit more difficult.
- L157: where does randomness arise if temperature is set to 0 (L150)
- Intuition lacking behind some design choices:
    - L266: what is the intuition behind defining the fitting information gain wrt as the reduction in uncertainty arising from $\theta$? In L269, why is the $\theta_k$ and $\theta_{k-1}$ used, and what do these mean? In L271, why is $J_r$ not defined mathematically in the approximation?
    - L275: what is the intuition again behind the definition of the compression penalty using $\theta_k$ and $\theta_{k-1}$?
- Weird citation: [26] is a generic citation that does little to explain how a Gaussian distribution is a mild assumption. You should explain how the CLT applies to model parameters.
- Please fix the legends in figure 2. (you can keep only one and put it next to the final figure to make the bars visible)
- Please fix Figure 4, it look like it should be a barplot. Otherwise, I would appreciate the intuition behind having a regular line plot.

---

> ### Author Rebuttal · Authors · 2025-07-29
>
> # Responses for Reviewer Sp46
>
> We sincerely appreciate the reviewer Sp46's constructive suggestions and the time and effort dedicated to the review process. We are also grateful for the recognition of our work, which is a great encouragement to us, and sincerely hope the following responses can help address the concerns.
>
> ## **Response to W1**
>
> We sincerely appreciate the valuable suggestions and have carefully polished the paper, including:
>
> * Removed redundant explanations of previously defined symbols (e.g., $B_{token}$ and $T_{max}$), defining them in detail only where they first appear.
> * Added a symbol table before Appendix A, listing all symbols and their corresponding meanings.
>
>
>
> ## **Response to W2**
>
> * In the motivation experiment, we use greedy decoding to focus on analyzing the impact of outcome reward RL on the trade-off between efficiency and effectiveness (L150), avoiding the bias and randomness introduced by sampling. However, considering that there may exist other sources of randomness, such as the model’s adaptive episode segmentation guided by prompts, where different truncation points may slightly affect the context and outputs, we still adopt the commonly used evaluation metric maj@4 (L157).
>
> * In contrast, for performance analyses, we do not use greedy decoding; instead, we set the temperature to 0.6 (L887), following the same experimental setup and evaluation protocol with baselines for comparison.
>
> * We appreciate for pointing out this insightful question and further emphasized the above clarification at L157.
>
>
>
> ## **Response to W3**
>
> We would like to provide an outline to convey the intuition behind our design:
>
> **For the fitting information gain:**
>
> - **How to interpret LLM reasoning from an information-theoretic perspective (fitting gain vs. uncertainty):** The reasoning process of an LLM can be viewed as inferring the correct answer $Y$ from input $X$. The more certain the model's prediction is, the better it "understands" the answer. Based on classical information theory [3, 25], we can use conditional entropy $H(Y|X)$ to characterize the model's uncertainty about the output $Y$. If the model is sufficiently confident, its $H(Y|X)$ should be low; conversely, if it is uncertain or making a blind guess, $H(Y|X)$ should be high. Within this framework, the goal of inference is to gradually reduce $H(Y|X)$ until the correct answer is output. Importantly, while $H(Y|X)$ represents the relationship between the input and output, this process is controlled by the LLM (determined by the parameter $\theta$). **Therefore, a more reasonable metric is: Given $\theta$, what is the model's uncertainty about $Y$?** That is, we want $\theta$ to not only capture the input $X$ but also provide strong discrimination of the correct answer $Y$. Therefore, we use conditional mutual information $I(\theta; Y | X) = H(Y|X) - H(Y|X, \theta)$ to measure how much the known model parameters $\theta$ reduce the uncertainty about $Y$ given $X$. This ties rewards to meaningful reasoning progress, i.e., the reward for each episode depends on how much it helps the model understand the correct answer. Therefore, we define "fitting information gain as the reduction in uncertainty".
>
> - **How to characterize the gradual improvement of inference (introducing episode gain with $\theta_k$ and $\theta_{k-1}$):** In each episode $k$, the model updates its parameter state from $\theta_{k-1}$ to $\theta_k$ by observing the new reasoning step. **The key question is: Does this episode help the model better "understand" the answer?** Therefore, we define fitting information gain of this episode as $\Delta I_k = I(\theta_k; Y | X) - I(\theta_{k-1}; Y | X)$. If this incremental gain is significant, it indicates that this episode has helped the model become more certain and effective. Here, $\theta_k$ and $\theta_{k-1}$ represent the posterior model parameters before and after learning the knowledge from episode $k$, estimated by the change in log-probability during the forward prediction process.
>
> - **Why the proposed metric corresponds to "reducing uncertainty" (approximating information gain with $J_r$):** Because directly calculating mutual information is too complex, we introduce an approximate metric $J_r(\cdot)$ to represent the model's predicted probability of the correct answer. This metric improves as the model's "confidence" increases. Therefore, the fitting gain is approximated as $\Delta I_k \approx J_r(\pi_\theta(\cdot \mid s_k, z_k)) - J_r(\pi_\theta(\cdot \mid s_k))$, with theoretical guarantees in Appendix C.1 and Proposition C.1. This difference measures whether episode $k$ improves the model's ability to predict the correct answer, reflecting the reduction process, i.e., the increasing reliability of reasoning. **As for the definition of $J_r$**, we briefly described it in L270 and detailed it in Appendix C.1. We sincerely apologize for the confusion caused by the location and have now added the precise formula and definition to the main text.
>
> **For the compression penalty:**
>
> - **Why introducing compression penalty:** In LLM reasoning, if each episode's information causes a significant change, while that episode may provide information gain, it may also capture unnecessary details within that episode, leading to overfitting or redundant computation. We want the model to only learn information that contributes to the answer within the episode. Therefore, inspired by the information bottleneck theory, we introduce a compression penalty based on the fitting term to further improve efficiency. It measures the "information overhead" incurred by each episode from an information-theoretic perspective.
> - **How is it measured (intuition behind the design):** If $\theta_k$ differs significantly from $\theta_{k-1}$, but the model's prediction performance (i.e., fitting gain) improves only slightly, this step may "absorb redundant information". Therefore, we use the mutual information increment $I(\theta_k; s_k) - I(\theta_{k-1}; s_{k-1})$ between them *with the fitting term* to measure whether the information in episode $k$ introduces unnecessary overhead.
> - **Why this is called compression:** The term *compression* comes from the idea that a model should retain only the **minimal amount of information sufficient** to perform accurate reasoning. In our setting, the mutual information $I(\theta; s_k)$ quantifies how much the information of this episode is encoded into the model parameters $\theta$. A larger value implies that the model has to “store” more bits to fit that episode, akin to using more storage in a compressed file. By penalizing the increase $I(\theta_k; s_k) - I(\theta_{k-1}; s_{k-1})$, we explicitly discourage storing redundant information, promoting a more compact (i.e., compressed) internal representation. This aligns with principles from MDL and PAC-Bayes, where generalization is favored when the hypothesis (here, $\theta$) is simple and concise.
>
> We have further added the above illustrations in Section 4.2 according to the valuable suggestions.
>
>
>
> ## **Response to W4**
>
> We appreciate the suggestions and have further added a justification for the Gaussian assumption in Appendix C.1.
>
> In brief, during LLM training, parameter updates $\theta$ can be viewed as the accumulation of numerous small stochastic gradient steps. Each step introduces a small, random perturbation in parameter space, effectively acting as the sum of many independent random variables. When these perturbations are sufficiently numerous and diverse, the overall distribution of parameter changes tends toward a Gaussian distribution, as implied by the Central Limit Theorem [26].
>
> Therefore, when estimating the compression term, we model the low-rank surrogate $\tilde{\theta}$ (obtained via SVD) using a Gaussian distribution. This assumption aligns with covariance approximation techniques in the PAC-Bayes framework and Fisher information-based methods, allowing us to derive a closed-form expression for mutual information and significantly reduce computational cost. Relevant details have been added to Appendix C.2, with a brief explanation in the main text.
>
>
>
> ## **Response to W5 & 6**
>
> We sincerely thank the constructive suggestions and have improved the figures accordingly:
>
> - The legend in Figure 2 has been moved below to ensure each bar is clearly visible.
> -  Figure 4 has been redrawn as a bar chart to better illustrate the ablation results.
>
>
>
> ## **Response to Question 1**
>
> We sincerely appreciate the reviewer for the valuable suggestions and are grateful for the time and effort devoted to the review. Based on this constructive feedback, we have improved the paper accordingly (as detailed above). We promise that new descriptions will appear in the paper as expectations ("as to what is needed to address the mentioned confusion").
>
>
>
> ## **Response to Limitations**
>
>
> We sincerely thank the valuable suggestions and further added the following discussion in Appendix C.3:
>
> - The token budget serves as an important constraint on test-time compute, helping to assess a model’s reasoning ability under limited compute. In our experiments, we adopt the same budget settings and evaluation protocols as the baselines to ensure fair and reproducible comparisons.
> - L2T does not rely on hard budget truncation. Instead, it uses a universal dense reward to guide the model to adaptively select the most informative tokens at each step, enabling dynamic adjustment of reasoning length. The model can flexibly determine the number of steps based on task complexity, converging early on simpler tasks or extending reasoning on more complex ones.
> - This provides a way for more explicit and automated dynamic budget allocation in the future, especially in scenarios sensitive to token costs (e.g., mobile deployment or real-time QA). We sincerely appreciate the constructive comments, where integrating reward signals with adaptive budget control is meaningful.

---

> > ### Comment · Reviewer_Sp46 · 2025-08-04
> >
> > Thank you for the response and the explanations. Please try to integrate some of these into the main text to help future readers.

---

> > > ### Author Response · Authors · 2025-08-04
> > >
> > > Dear Reviewer Sp46:
> > >
> > > We sincerely appreciate your time, effort, and constructive comments on the review and discussion. We have integrated the mentioned descriptions into the paper according to the valuable suggestions.
> > >
> > > Best wishes,
> > >
> > > The authors.

---

### Official Review · Reviewer_fdAZ · 2025-07-01

**Clarity:** 3
**Significance:** 3
**Originality:** 3
**Rating:** 5
**Confidence:** 3

**Summary:**

This paper proposes L2T, which uses an information-theoretic dense reward for RL training of reasoning models. The core idea is to estimate information gain by comparing the probability score of the answer with and without the corresponding reasoning episode.

**Questions:**

See the weakness for more questions

- How are the generated reasoning chains segmented, and how do you control the number of episodes (up to 30)?
- How do you break the reasoning chains into *K* segments?
- Why do you think outcome-reward RL does not lead to a significant accuracy improvement?
- What would happen if L2T were applied to non-reasoning models like Qwen1.5?

**Ethical Concerns:**

["NO or VERY MINOR ethics concerns only"]

**Final Justification:**

As outlined in my review, the method makes sense and also the author rebuttal addresses most concerns, although there are some presentation issues. Therefore, I’ve raised my score to 5.

**Limitations:**

The authors stated that they discussed the limitation in the checklist, but I couldn’t find it.

**Quality:**

3

**Strengths And Weaknesses:**

**Strengths:**

- The proposed solution is interesting. Directly using the informativeness of each reasoning step as a process reward makes sense.
- The method is general and could be applied to various forms.
- The paper is written with a clear narrative and is easy to follow, although some parts feel unnecessarily long.

**Weaknesses:**

- There are many existing works addressing overthinking and the elimination of redundant steps. The paper should include this line of research on concise reasoning and consider adding them as baselines. [1]
- Continuing from that, the motivation is already well-addressed in prior work. Since the paper makes a solid contribution, I suggest trimming the motivation and instead adding more extensive evaluations, such as on coding tasks.
- The exact implementation details for the low-rank approximation are unclear; the explanation is too verbose and lacks clarity. Efficiency aspects are also not well discussed.
- The baseline scores are much lower than those reported in the original papers (DeepSeek and DeepScaler). Is there a reason for this discrepancy?
- There is no qualitative analysis. I’m curious how the method eliminates redundant reasoning at low vs. high alpha values.
- There are several typos (e.g., effectiiveness) and unclear phrases (e.g., “both models” in line 163, while only one result is shown in the main paper).

**Reference:**

[1] Chen et al., *Do NOT Think That Much for 2+3=? On the Overthinking of o1-Like LLMs*, 2024

---

> ### Author Rebuttal · Authors · 2025-07-29
>
> # Responses for Reviewer fdAZ
>
> We sincerely appreciate the reviewer fdAZ's constructive suggestions and the time and effort dedicated to the review process. We are also grateful for the recognition of our work and sincerely hope the following responses can help address the concerns.
>
> ## **Response to Weaknesses 1 & 2**
>
> - We would like to kindly clarify that in the original paper, we have cited [1] as [9]. As suggested, we further added more discussion and comparison with [1] in Sections 2 and 5:
>   - While both our method and [1] aim to reduce unnecessary reasoning in LLMs, our goals and mechanisms differ. [1] mainly focuses on early stopping effects, aiming to detect when the model has already produced the correct answer and can halt generation. It may still allow inefficient reasoning traces before the correct answer appears. In contrast, our method proactively maximizes per-step information gain during training, encouraging the model to only generate reasoning steps that are most useful, without relying on any external stopping criterion.
>   - We are supplementing the comparison with the baseline [1] using the same settings (currently running). Due to time constraints for the rebuttal, we will add the comparative experimental results in the final version.
> - We have compared L2T with multiple baselines targeting "overthinking" and "efficiency" in Sections 5, e.g., [2, 5,36,62]. As shown in Tables 1–2, L2T achieves nearly 2% performance gain while using less than 70% of the tokens, outperforming prior SOTA methods.
> - We sincerely appreciate the reviewer's constructive suggestions and affirmation of our work. According to the comments, we further refined the motivation section and moved the experimental results on coding tasks from Appendix F.2 to the main text (Figure 8).
>
>
>
> ## **Response to Weakness 3**
>
> **Implementation for low-rank approximation**: The goal is to obtain a low-rank approximation $\tilde{\theta} \in \mathbb{R}^r$ of the model parameters $\theta \in \mathbb{R}^d$ ($r \ll d$), avoiding direct computation of the Fisher matrix in high-dimensional space (Appendix C.1 and L288–299). Specifically, we apply SVD to extract the principal directions of variation in $\theta$ and retain the top $r$ components (with $r/d \approx 1%$–10% for 1.5B and $r/d \approx 0.1%$–1% for 7B), resulting in a low-rank surrogate $\tilde{\theta}$.
>
> **Improvement**: Building on Appendix C.1, we further incorporated the above explanation into the main text and revised the writing to improve clarity, following the valuable suggestions.
>
> **Efficiency**:
>
> * **For the mentioned low-rank approximation:**
>   * **Method:** This mechanism avoids directly calculating the Fisher matrix in high-dimensional space, and instead uses low-rank proxy calculations to reduce the amount of computation and improve efficiency.
>   * **Evidence:** Appendix C.1 theoretically shows that the method reduces complexity from $\mathcal{O}(d^2)$ to $\mathcal{O}(dr + r^2)$; Section 5.3 and Appendix F.3 (configurations ii and iii) empirically demonstrate that it introduces minimal computational overhead while improving performance (Figures 4 and 9).
> * **For L2T:**
>   * **Method:** L2T introduces a general information-theoretic dense process reward to guide the model toward reasoning steps that yield the most performance gains. By assigning higher rewards to informative steps and down-weighting redundant ones, the model learns to recognize effective reasoning. Then, the policy optimization based on this reward encourages the model to make the most of each step, generating accurate answers with minimal token usage.
>   * **Evidence:** The comparison results in Section 5.2 and Appendix F.2 show that L2T triples LLM's token efficiency (Figure 2-3) and improves accuracy by more than 2% (Table 1-2) across various tasks.
>
> We appreciate the constructive suggestions and have further polished the text and emphasized the above analysis in the main text as suggested.
>
>
> ## **Response to Weakness 4**
>
> This discrepancy may stem from differences in experimental settings and evaluation protocols. We follow the setup of [36] for our evaluations, and the scores of all baselines are consistent with those reported in [36] (Section 5.1 and Appendix E).
>
>
>
> ## **Response to Weakness 5**
>
> Following the valuable suggestions, we further added a qualitative analysis for $\alpha$.
>
> The coefficient $\alpha$ denotes the weight of our proposed process reward (Eq.5). When $\alpha$ is small, the model relies more on the outcome reward, which provides high rewards only when the correct answer is found. Without guidance and given that correct answers are sparse in the output space, the model may consume a large number of tokens in exploration, reducing efficiency. In contrast, when $\alpha$ is large, the model is driven by the process reward, which assigns high rewards only if the current reasoning step has a high contribution to the accuracy of the answer, i.e., the correct answer is reached within a short token sequence. This encourages the model to generate informative tokens at each step, thereby improving efficiency.
>
> Take the Tier-1 Omni-MATH problem “What is 12 + 5?” as an example, the following qualitative analysis shows that with $\alpha = 0.9$, the model may answer within 2–3 steps, whereas with $\alpha = 0.2$, it may take more than 7 steps. More cases have been included in Appendix F.4.
>
> ```
> - alpha=0.9:
> 12 plus 5 equals 17.
> Answer: 17
>
>
> - alpha=0.2：
> To solve this, we first recall the addition operation.
> Addition is combining two numbers into a total.
> Let’s start with 12 and add 5.
> 12 plus 5 equals 17.
> We double-check: 10 + 5 = 15, so 12 + 5 = 17.
> The final result should be 17.
> Answer: 17
> ```
>
> ## **Response to Weakness 6**
>
> We sincerely appreciate the reviewer's suggestion and have made the corresponding revisions, including:
> - correcting “effectiiveness” to “effectiveness” in L186;
> - changing “Figure 1” to “Figures 1 and 6” in L162.
>
>
>
> ## **Response to Questions 1 & 2**
>
> We automatically segment the chains by designing specific prompts. Taking mathematical reasoning tasks as an example:
>
> For the motivation experiment ("segment up to 30" in Question 1), we added the following instruction to the prompt file:
>
> ```
> <think>
> In this section, show your detailed reasoning process.  Break down your reasoning into at most 30 logically coherent segments.
>
> Each segment must be clearly marked with numbered tags in the format <episode_1> ... </episode_1>, <episode_2> ... </episode_2>, ..., up to <episode_30>.
>
> Ensure each <episode_i> should contain only a single complete logical move, such as a definition, a formula derivation, a transformation, or a case split.
> ...
> </think>
> ```
>
> For performance analysis ("break the reasoning chains into K segments" in Question 2), we added:
>
> ```
> <think>
> In this section, show your detailed reasoning process.  Break down your reasoning into logically coheret segments.
>
> Each segment should be enclosed in <episode_k> ... </episode_k> tags, such as:
> <episode_1>...</episode_1>
> <episode_2>...</episode_2>
> ...
> <episode_K>...</episode_K>
>
> Ensure each  <episode_i> should contain only a single complete logical move, such as a definition, a formula derivation, a transformation, or a case split. Avoid grouping multiple logical steps into one.
>
> ...
> </think>
> ```
>
>
>
> ## **Response to Question 3**
>
> We would like to kindly clarify that while we acknowledge the effectiveness and potential of outcome-reward RL, its improvements may be limited *under constrained token budgets*, and in some cases, performance degradation could occur.
>
> Specifically, when the model relies solely on outcome rewards, high rewards are only given upon reaching the correct answer. Without intermediate guidance and given the sparsity of correct answers in the output space, the model may consume a substantial number of tokens during exploration, potentially affecting efficiency (L31–47).
>
> The motivation experiment in Section 3.2 and performance analyses in Section 5 further demonstrates this. Under limited budgets, outcome-reward RL may introduce unnecessary reasoning in simpler tasks, which could hinder decision-making, or prematurely exhaust tokens in complex tasks, risking the truncation of critical steps and a drop in overall performance.
>
>
>
> ## **Response to Question 4**
>
> When applied to the mentioned "non-reasoning models like Qwen1.5", L2T can still yield performance improvements, as supported by the following rationale and evidence:
>
> 1. **Methodology**: L2T does not rely on strong inherent reasoning abilities of the base models; even for models with limited reasoning capabilities, L2T can still be effective by using heuristically segmented episodes and learning from dense feedback during training. Moreover, L2T operates based on the model’s internal probability outputs and parameter dynamics, without requiring external evaluators, making it applicable to a wide range of LLMs.
>
> 2. **Empirical Evidence**: Appendix F.2 presents the experimental results on Qwen2-7B-Instruct. Similar to the mentioned Qwen1.5, Qwen2-7B-Instruct has not been fine-tuned specifically for reasoning tasks; its reasoning ability largely stems from broad coverage in pretraining data [1,2]. As shown in Figure 8, compared to GRPO, applying L2T leads to over a 4% improvement on code reasoning tasks.
>
> [1] Qwen Technical Report.
>
> [2] Qwen2 Technical Report.
>
>
>
> ## **Response to Limitations**
>
> We would like to kindly clarify that we have discussed the limitation in Appendix C.3, and sincerely apologize for any concern it may have caused by the location. We have now moved the corresponding discussion in Appendix C.3 to the end of Section 6 in the main text.

---

> > ### Comment · Reviewer_fdAZ · 2025-08-06
> > **Response to the authors**
> >
> > Thanks for the detailed rebuttal, and apologies for the delayed response due to some eye issues.
> >
> > After reading through everything, I believe there are no misunderstandings on my side regarding the paper. On the presentation side, I think the paper should be evaluated based on the submitted version.
> >
> > From a research perspective, I find the intuition and method both sensible and novel. Although the mechanisms differ somewhat, as pointed out in the original weaknesses, it would be beneficial to include more comparisons on that aspect. Personally, I believe that even simple SFT with careful rejection sampling could reduce token usage and improve performance, if trained solely on correct mathematical reasoning.
> >
> > Considering all of this, I will maintain my score of borderline accept.

---

> > > ### Author Response · Authors · 2025-08-06
> > >
> > > Dear Reviewer fdAZ,
> > >
> > > We sincerely thank you for your positive feedback and for recognizing the novelty and significance of our work. It is a great honor that we were able to address all your concerns during the discussion. Based on your valuable suggestions, we have carefully revised the manuscript and incorporated all the necessary modifications, sincerely hoping these additions can help further consider the contributions and practical relevance of our work. Below, we would like to take this opportunity to further clarify the raised two points based on the original submission.
> > >
> > > Regarding your insightful comment on SFT and rejection sampling for reducing token redundancy, we fully agree that these methods can be effective when trained on correct reasoning trajectories. However, under strict token budget constraints, solely relying on early stopping or rejection sampling may offer limited performance gains. For complex reasoning tasks, models might prematurely deplete their token budget on sub-optimal steps, risking the omission of critical reasoning components. In contrast, our proposed method, L2T, explicitly optimizes the dense information gain at each reasoning step. Only steps that significantly contribute to the final answer receive high rewards, while redundant steps are penalized. This targeted optimization guides the model to prioritize tokens with the greatest informational value, ensuring efficient token utilization and preserving key reasoning steps even under budget constraints. Thus, L2T enhances the effectiveness of each generated token, rather than relying on output truncation.
> > >
> > > Furthermore, as detailed in our original submission, we have conducted extensive comparisons with multiple works focused on "token usage", e.g., ReST-MCTS [62] (Nov 2024), length penalty [2] (Feb 2025), and MRT [36] (Mar 2025). Across these baselines, our method consistently demonstrates superior reasoning efficiency and accuracy. We also appreciate your suggestions regarding additional comparisons, which have been incorporated into the revised paper.
> > >
> > > Finally, we would like to express our gratitude again for your constructive feedback and deeply appreciate the time and effort you devoted to reviewing and discussing our paper.
> > >
> > > Best regards,
> > >
> > > The Authors

---

### Public Comment · ~Jinyeong_Yun1 · 2026-06-01
**Question on the implementation of $J_r$ (Fitting Information Gain)**

Thank you for this interesting work. I have one question regarding the computation of $J_r$ in practice.

The paper states that the fitting information gain $\Delta I_k \approx J_r(\pi_\theta(\cdot|s_k, z_k)) - J_r(\pi_\theta(\cdot|s_k))$ requires only two forward passes of $\pi_\theta$ without gradient updates, relying solely on internal model signals. Given that the experimental benchmarks (AIME, AMC, MATH500) predominantly involve single-token numeric answers, the most natural interpretation seems to be $J_r(\pi_\theta(\cdot|s_k)) = \pi_\theta(y^*|s_k)$, i.e., the probability assigned to the ground-truth answer token.

However, the released code computes this via an external 7B PRM (Qwen2.5-Math-PRM-7B), which appears to contradict the paper's claim of using only internal signals without task-specific evaluators.

Could the authors clarify what implementation of $J_r$ was actually used in the reported experiments?

---

### Decision · Program_Chairs · 2025-09-17

**Decision:**

Accept (poster)

**Comment:**

This submission proposes Learning to Think (L2T). It is an information-theoretic reinforcement fine-tuning framework with the goal of improving the reasoning efficiency and effectiveness of llms. Specifically, in L2T the authors introduce a dense process reward that estimates episode-wise information gain based on PAC-Bayes bounds and the Fisher information matrix. In doing so, it requires no additional annotations or task-specific evaluators. Experiments on reasoning benchmarks show the proposed method offers improvements over standard outcome-based RL methods.

Strengths:

1. dense reward design: the information-theoretic approach to defining process rewards is well-motivated and avoids the need for annotated reasoning traces, making the method general and broadly applicable.
2. improved reasoning efficiency: experiments demonstrate that L2T boosts both reasoning accuracy and efficiency across multiple benchmarks, outperforming existing outcome-reward RL baselines.
3. theoretical grounding: the use of PAC-Bayes bounds and Fisher information provides a sound theoretical basis for the reward estimation.

Weaknesses that are raised by the reviewers:

1. incomplete evaluation: while the results are promising, several reviewers noted the lack of ablation studies, scaling experiments on larger models, and comparisons with more related works addressing concise reasoning and RL-based efficiency improvements.
2. clarity and presentation: sections with dense formulas are difficult to follow, and the lack of a clear schematic diagram for the overall framework makes it harder for readers to grasp the main ideas.
3. limited qualitative insights: the submission lacks qualitative analyses demonstrating how redundant reasoning is reduced and how reasoning depth adapts under different reward weights.

overall, this submission presents a technically sound and well-motivated method that addresses an important challenge in improving LLM reasoning efficiency. While some evaluations and presentation aspects could be strengthened, the proposed approach is theoretically sound, practical, and demonstrates consistent empirical gains. Most importantly, the authors managed to address many of the issues during the rebuttal phase, thus two reviewers did raised their ratings. Given the four positive ratings from four reviewers and the addressed concerns during rebuttal, an acceptance recommendation is given to this submission.